

# Grammatical error correction for low-resource languages: a review of challenges, strategies, computational and future directions

Syauqie Muhammad Marier[1,2], Xiangfan Chen[1], Linan Zhu[1] and Xiangjie Kong[1]

[1] Zhejiang University of Technology, Hangzhou, Zhejiang, China
[2] Universitas Nahdlatul Ulama Yogyakarta, Sleman, Yogyakarta, Indonesia

Corresponding author
Xiangjie Kong, xjkong@ieee.org

## ABSTRACT

Grammatical error correction (GEC) is crucial for enhancing the readability and comprehension of texts, particularly in improving text quality in low-resource languages. However, challenges such as data scarcity, linguistic diversity, and limited computational resources hinder advancements in this domain. To address these challenges, researchers have developed strategies such as synthetic data generation, multilingual pre-trained models, and cross-lingual transfer learning. This review synthesizes findings from key studies to explore effective GEC methods for low-resource languages, emphasizing approaches for handling limited annotated corpora, typological complexities, and evaluation challenges. Synthetic data generation techniques, including noise injection, adversarial error generation, and translationese-based augmentation, have proven vital for overcoming data scarcity. Multilingual and transfer learning approaches demonstrate effectiveness in adapting knowledge from high-resource languages to low-resource settings, especially when combined with fine-tuning on curated datasets. Additionally, linguistic diversity has been partially addressed through methods like morphology-aware embeddings, byte-level tokenization, and contextual data preprocessing. However, limited research exists on robust evaluation metrics tailored to diverse typologies, such as agglutinative and morphologically rich languages, and the creation of gold-standard datasets remains an ongoing challenge. Recent advancements in dataset construction and the use of large language models further enrich this field, offering scalable solutions for low-resource contexts. Despite notable progress, this review identifies gaps in evaluation methodologies and typology-specific solutions, calling for future innovations in multilingual modeling, dataset creation, and computationally efficient GEC systems tailored to the unique needs of low-resource languages.

## INTRODUCTION

Grammatical error correction (GEC) is a critical task that focuses on carefully identifying and correcting various language errors present in a given text. GEC has broad applications,

from improving the writing of language learners to helping professionals generate error-free text. Although significant progress has been made in the development of GEC for high-resource languages such as English and Chinese (*Solyman et al., 2022*; *Flachs, Stahlberg & Kumar, 2021*; *Pajak & Pajak, 2022*), where large annotated datasets and powerful linguistic resources are available, scaling GEC systems to low-resource languages poses unique challenges. The scarcity of data, the presence of linguistic diversity, and typological differences have been identified as factors that create obstacles that conventional GEC methods often fail to overcome.

Most existing GEC systems are heavily relying on extensive data for training (*Sjöblom, Creutz & Vahtola, 2021*). These datasets are often unavailable or severely limited for low-resource languages, making it difficult to build comparable GEC systems. Furthermore, low-resource languages often exhibit significant linguistic diversity, including complex grammatical structures, rich morphologies, and varying orthographic conventions, requiring unique adaptations in tokenization, embeddings, model architectures, and evaluation protocols. Despite growing interest in addressing these challenges, no comprehensive review has yet been conducted to summarize the progress made and identify effective methods for GEC in low-resource settings.

Previous GEC surveys (*Wang et al., 2021*) and (*Bryant et al., 2023*) have provided comprehensive overviews of datasets, techniques, and challenges, but have focused mainly on high-resource languages, highlighting existing methods, data augmentation techniques, and evaluation approaches. However, to our knowledge, no previous surveys have systematically reviewed the challenges and methodologies specific to low-resource languages. The primary goal of this survey is to examine this gap by reviewing the current GEC approaches, particularly for low-resource languages. We focus on answering the following questions: '**Which strategies prove to be the most effective for GEC in low-resource languages**', and '**how can these strategies be tailored to effectively address the challenges of data scarcity and linguistic diversity?**' By identifying and analyzing key trends in data augmentation, transfer learning, multilingual GEC systems, and tokenization/embedding techniques, this survey aims to highlight effective strategies for low-resource language correction. Furthermore, we explore existing datasets and evaluation methods, highlighting how they can be expanded or adapted to better support GEC efforts in low-resource languages.

This survey offers the following key contributions.

- **Comprehensive analysis of effective strategies for low-resource GEC**: We thoroughly examine which strategies are most effective for GEC in low-resource languages and how these approaches can be tailored to address the dual challenges of data scarcity and linguistic diversity. Our analysis reveals that synthetic data generation, cross-lingual transfer learning, and hybrid modeling approaches offer the most promising solutions when adapted to the specific linguistic features of target languages.
- **Structured analysis of GEC methods for low-resource languages**: We present and evaluate methods such as synthetic data generation, cross-lingual transfer techniques, and hybrid modeling approaches, highlighting their strengths and limitations.

- **Detailed analysis of GEC system**: We present a thorough analysis of GEC computational techniques, categorized into four types: Transformer-based models, pre-trained multilingual models, adversarial approaches, and LLM-based methods. This analysis is supported by a comprehensive cross-linguistic performance comparison that highlights the effectiveness of each approach across various language typologies.
- **Comparison of datasets and evaluation metrics**: We offer a comprehensive overview of existing GEC datasets and evaluation protocols, highlighting their relevance for low-resource languages and suggesting necessary adaptations for typological diversity.
- **Challenges and future directions**: We identify key challenges that continue to affect in this field, including addressing linguistic diversity in multilingual systems, designing tokenization methods suitable for morphologically rich languages, and developing scalable evaluation metrics. Based on this review, we propose actionable recommendations to advance GEC research for low-resource languages.

This review is intended for researchers, academics, and practitioners in the fields of natural language processing (NLP), computational linguistics, and artificial intelligence who are interested in advancing GEC systems for low-resource languages. It is particularly valuable for those working on machine learning models, multilingual NLP, and linguistic resource development.

The remainder of this survey is organized as follows. First, we present the Survey Methodology, outlining the search strategies, databases used, and inclusion/exclusion criteria to ensure comprehensive and unbiased coverage of the literature. Then, we review the challenges specific to low-resource language GEC and contextualize them using background literature. The second section discusses methods for handling data scarcity, including synthetic data and cross-lingual strategies. In the following section, we explore computational techniques, including transformer-based models and adversarial approaches. We then review the evaluation metrics and their adaptations for low-resource GEC. This is followed by a discussion of existing datasets, tokenization methods, and embedding techniques. Finally, we highlight gaps in the literature and propose directions for future research, including improvements in synthetic data generation, multilingual models, and evaluation frameworks. The survey concludes with a summary of key findings.

## SEARCH METHODOLOGY

To compile this review, we conducted a thorough literature search to ensure complete coverage and avoid bias in the selection of articles. The search process involved the following steps:

### Search

We used several leading academic search engines and electronic databases to search for articles relevant to this research topic. The search engines used included: Google Scholar, IEEE Xplore, Scopus, ACM Digital Library, ScienceDirect, and SpringerLink. We used a combination of keywords aligned with the core themes of this review, ensuring relevance and breadth in the search results. Keywords such as 'Grammatical', 'Error', and

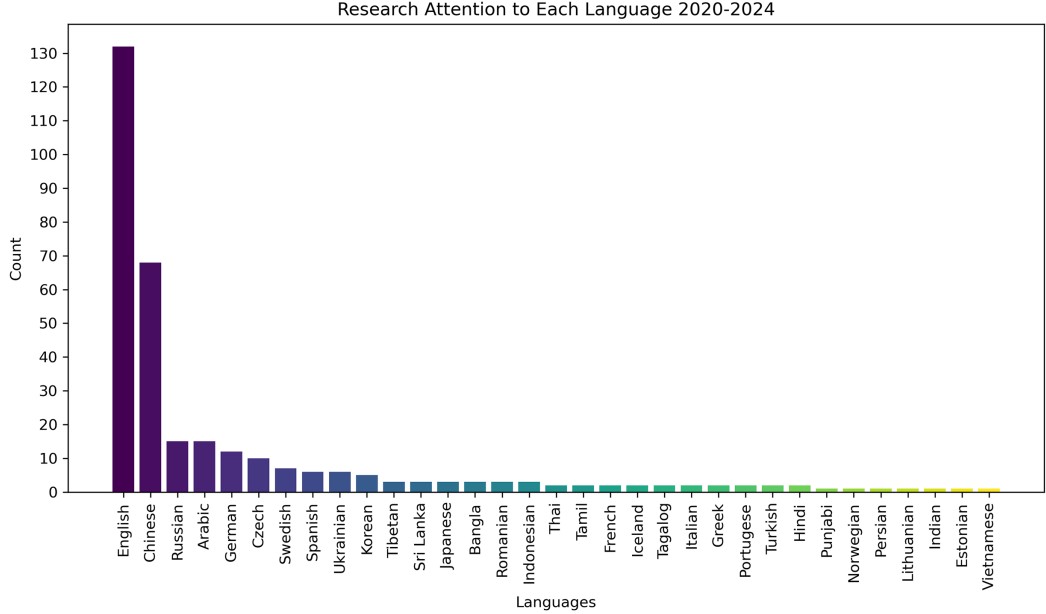

**Figure 1 Distribution of research attention across languages (2020–2024).**

'Correction' were used to collect relevant information. The selected keywords are central to the field and are commonly used in related research, ensuring alignment with the existing literature. To ensure a comprehensive search, variations and related terms were also considered, including spelling errors, low-resource, and GEC low-resource language. The research articles were collected between the years 2020 and 2024.

## Inclusion and exclusion criteria

We collected all relevant articles, including those discussing the creation of a GEC *corpus*, the development of GEC models and the formulation of robust strategies to improve GEC. Papers on the following topics were excluded:

- Grammatical errors in speech recognition tasks.
- Articles that analyze grammatical errors purely from a linguistic perspective.
- Duplicate research articles published in different journals.

To prioritize underrepresented languages and address the research gap in low-resource settings, we excluded studies focusing on English and Chinese, as these languages dominate the existing GEC literature. This decision was based on our analysis of the research trends in GEC, which showed that these two languages receive significantly more attention than others. As illustrated in Fig. 1, there is a pronounced disparity in research focus, with English and Chinese receiving disproportionately more attention than other languages.

To analyze these trends and compare the findings across different languages in the GEC research, we employed both quantitative and qualitative methods. Quantitatively, we

classified studies based on the languages addressed and the GEC models employed, ensuring a balanced representation of research contributions across different linguistic contexts. For example, a study that presents GEC models in both English and German, which would be counted in both language categories, ensuring that research contributions across different languages are accurately represented. This method helps us understand the level of research attention given to each languages. Qualitatively, we performed a thematic analysis to uncover recurring challenges, innovative solutions, and computational approaches specific to low-resource languages. The detailed results of this analysis will be discussed in the next section.

### Article selection and review

Following the initial search, articles were filtered based on their titles, abstracts, and relevance to the inclusion criteria. The relevant articles were subjected to a rigorous evaluation to assess their methodological quality, novelty, and potential to advance the understanding of GEC in low-resource languages. Only studies deemed to make a significant contribution were included in this review.

The search methodology was systematically applied to ensure a comprehensive and unbiased selection of the literature, providing a comprehensive overview of the subject matter while minimizing potential bias in the inclusion of studies.

## BACKGROUND AND KEY CONCEPTS

### Low-resource languages

Low-resource languages are those that lack sufficient computational, linguistic, and annotated resources, such as corpora, tools (*e.g.*, tokenizers and parsers), or pre-trained models that are specifically designed for these languages (*Hedderich et al., 2021*). These languages include Zarma (*Keita et al., 2024*), Modern Greek (*Korre & Pavlopoulos, 2022*), Icelandic (*Ingólfsdóttir et al., 2023*), Tagalog (*Lin et al., 2023*), Tamil (*Sakuntharaj & Mahesan, 2021*), Icelandic (*Ingólfsdóttir et al., 2023*) and Arabic (*Belkebir & Habash, 2021*). These languages often exhibit complex linguistic features, such as agglutination, flexible word order, and tonal systems, that hinder their processing in NLP tasks (*Belkebir & Habash, 2021*; *Uz & Eryiğit, 2023*; *Korre, Chatzipanagiotou & Pavlopoulos, 2021*). These challenges are further compounded by the scarcity of high-quality datasets and language processing tools, as most open-source NLP pipelines are primarily designed for resource-rich languages like English (*Sun et al., 2022*; *Luhtaru, Korotkova & Fishel, 2024*; *Musyafa et al., 2022*). Furthermore, the lack of annotated corpora restricts the use of supervised learning, as many available datasets consist of noisy or small-scale learner data (*Náplava & Straka, 2019*; *Flachs, Stahlberg & Kumar, 2021*). The absence of pre-trained models for these languages further exacerbates the challenges in tasks such as GEC (*Ingólfsdóttir et al., 2023*; *Korre & Pavlopoulos, 2022*).

### Challenges in grammatical error correction

The primary challenge in GEC for low-resource languages is the scarcity of annotated corpora. Although languages like English benefit from large datasets (*e.g.*, Lang8, CoNLL,

JFLEG), low-resource languages often lack such resources, forcing researchers to rely on noisy data or synthetically generated error-annotated corpora (*Náplava & Straka, 2019*; *Flachs, Stahlberg & Kumar, 2021*). This scarcity, combined with three major obstacles, complicates the development of effective GEC systems.

- **Linguistic characteristics**: Agglutination and free word order in languages such as Turkish (*Uz & Eryiğit, 2023*), as well as the complex morphology of Arabic (*Belkebir & Habash, 2021*), introduce unique errors that require customized error classification systems.
- **Data scarcity**: Low-resource languages lack sufficient annotated corpora for training GEC systems. To address this, methods such as error injection, back-translation, and multilingual transfer learning are increasingly utilized to generate synthetic data and leverage high-resource models.
- **Limitations of existing evaluation metrics**: Current evaluation metrics like GLEU and BLEU are not ideal for capturing nuanced errors in morphologically rich or syntactically flexible languages. ERRANT and other distance metrics for linguistic edit also struggle with complex error types, such as agreement or compounding errors (*Fang et al., 2023a*).

For example, in agglutinative languages such as Turkish, errors such as 'gidiyirum' instead of 'gidiyorum' highlight the need for systems that can handle verb inflection errors. Similarly, the presence of orthographic ambiguities and syntactic flexibility in Arabic complicates the parsing process, demonstrating the need for language-specific GEC strategies. The variable noun cases and verb forms of Russian (*Rozovskaya, 2022*), the intricate word spacing and morphological rules of Korea (*Yoon et al., 2023*), and the numerous inflections of modern Greek and the flexible word order each present a unique challenge (*Korre, Chatzipanagiotou & Pavlopoulos, 2021*). These challenges underscore the need for creating error correction systems that are sensitive to linguistic diversity.

Data scarcity presents a fundamental challenge in developing GEC systems designed for languages with limited resources. Advanced GEC systems often utilize Transformer architectures, especially sequence-to-sequence models, which require large amounts of annotated data for training. This dependency limits their applicability to languages with scarce resources. To address this, key approaches include synthetic data generation methods, multilingual and cross-lingual learning, and efforts to construct datasets for low-resource languages including using advanced tools such as GPT-3/4 and manual curation. These strategies mitigate the lack of annotated corpora by leveraging high-resource language models and linguistic knowledge. In Handling Data Scarcity in Low-Resource Languages, we explore each of these techniques in detail.

Evaluation metrics such as ERRANT and GLEU often struggle to handle high morphological variation or syntactic changes in low-resource languages. For example, ERRANT has difficulty identifying errors in verb inflection or word composition in languages such as Modern Greek or Finnish (*Korre & Pavlopoulos, 2022*). However, metrics like ERRANT and F-score heavily rely on high-quality reference corpora, which are often unavailable for low-resource languages. As a result, noisy or synthetic corpora

can introduce biases, while weak supervision or crowdsourcing rarely aligns with the structured scoring frameworks these metrics require (*Keita et al., 2024*).

Metrics such as BLEU and ERRANT emphasize token-level differences over meaning preservation, limiting their effectiveness for agglutinative languages like Turkish or Hungarian (*Luhtaru, Korotkova & Fishel, 2024*; *Korre & Pavlopoulos, 2022*). This focus often overlooks broader semantic correctness, especially in languages where subword alignment poses challenges. Most metrics, including BLEU and ERRANT, were designed for monolingual tasks involving languages such as English and fail to consider the structural and grammatical features of multilingual or code-switched GEC tasks (*Flachs, Stahlberg & Kumar, 2021*).

## Overview of grammatical error correction

Most GEC systems rely on neural methods, particularly sequence-to-sequence (seq2seq) transformers, which effectively model grammatical corrections by learning patterns from large datasets. For example, transformer architectures with pretraining and fine-tuning in noisy–corrected text pairs have been widely used for high-resource languages and adapted to data-scarce settings through techniques such as unsupervised pretraining (*Grundkiewicz, Junczys-Dowmunt & Heafield, 2019*). Hybrid approaches, which combine rule-based techniques with neural methods, have proven effective in low-resource contexts, as they leverage explicit linguistic knowledge to compensate for data gaps (*Korre & Pavlopoulos, 2022*; *Keita et al., 2024*). In the section Computational Techniques and Multilingual Models, we present techniques: Transformer-based, multilingual pre-trained models, adversarial approaches and GANs, LLM-based, and Legacy approaches. On the evaluation side, commonly used metrics include GLEU and ERRANT, though their limitations in handling typological diversity and compound-rich languages are frequently noted (*Alhafni et al., 2023*; *Korre & Pavlopoulos, 2022*). Although precision-focused metrics such as $F_{0.5}$ are often preferred in GEC tasks, they may underrepresent rarer but linguistically meaningful correction types.

## Emerging solutions

Emerging solutions to address GEC challenges in low-resource languages include synthetic data generation, multilingual pre-training and cross-lingual transfer learning. Synthetic data generation, such as noise injection or adversarial error creation, has proven effective for augmenting annotated datasets (*Mahmoud et al., 2023*; *Katsumata & Komachi, 2019*; *Grundkiewicz, Junczys-Dowmunt & Heafield, 2019*).

Cross-lingual learning uses common characteristics found in both high-resource and low-resource languages, using models developed on extensive high-resource datasets and then fine-tuned with data from the target low-resource language, resulting in improved GEC performance (*Sun et al., 2022*; *Yamashita et al., 2020*). Pre-trained multilingual models, such as mT5 and Llama-based language models, are increasingly used for their ability to generalize across various languages with limited data (*Luhtaru, Korotkova & Fishel, 2024*; *Luhtaru et al., 2024*). Additionally, large language models (LLMs) like GPT-3.5, GPT-4 and Llama have shown promise in few-shot and

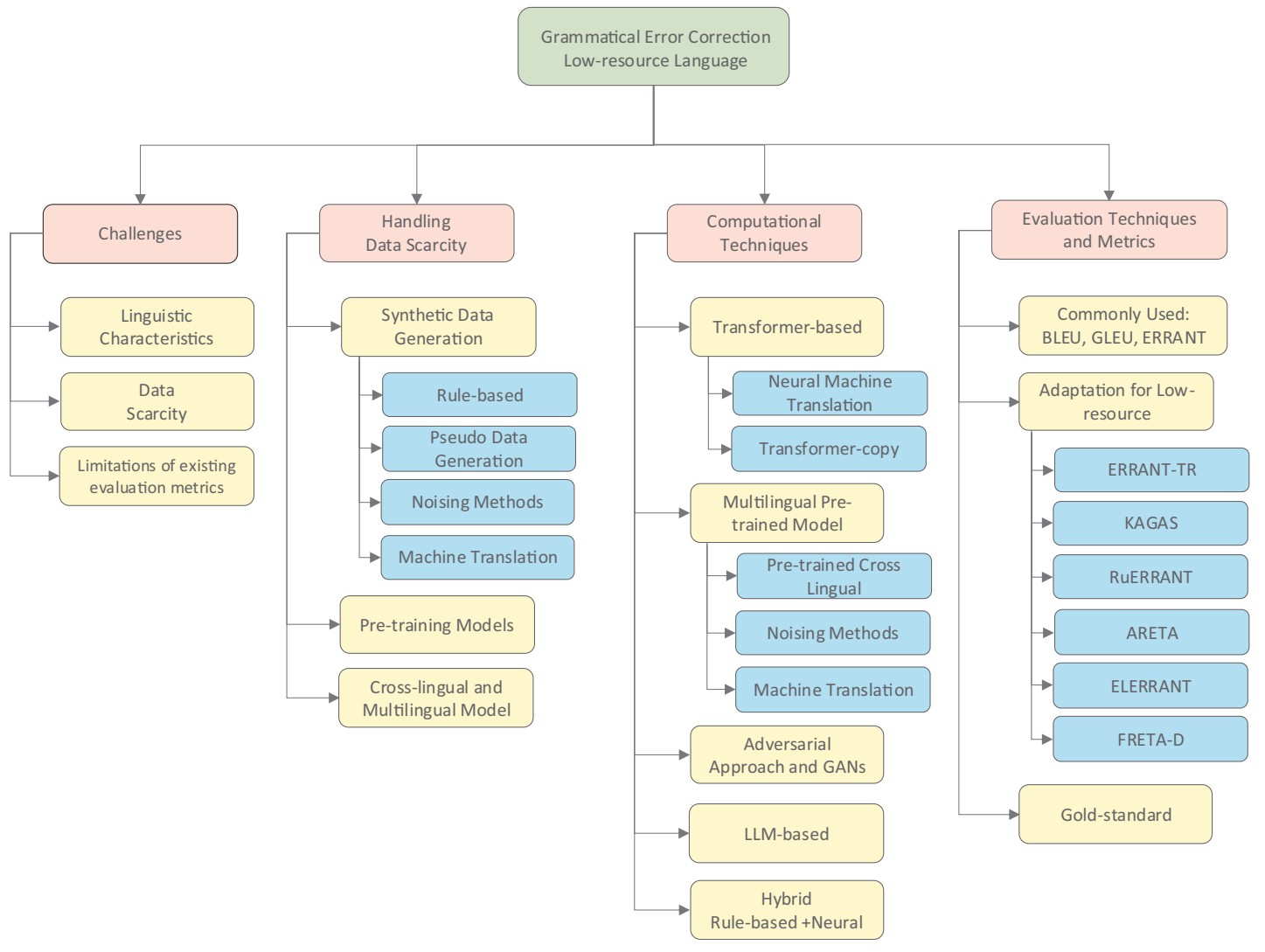

**Figure 2  The taxonomy of GEC low-resource language approaches discussed.**

instruction-tuned scenarios, further bridging resource gaps (*Luhtaru et al., 2024*). Together, these solutions represent a growing toolkit for tackling the persistent challenges of GEC in low-resource languages.

Figure 2 provides a comprehensive taxonomy of GEC approaches for low-resource languages discussed in this survey. This visual framework categorizes the methods into core dimensions: handling data scarcity, addressing linguistic diversity, computational techniques, and evaluation approaches. As illustrated, researchers have developed various strategies to overcome primary challenges, including synthetic data generation to address limited corpora, linguistic adaptations to handle diverse typologies, and specialized evaluation metrics tailored to different language structures. This taxonomy serves as a conceptual roadmap for understanding the relationships between different approaches discussed throughout this review.

# HANDLING DATA SCARCITY IN LOW-RESOURCE LANGUAGES

## Data generation and augmentation strategies

Grammatical error correction for low-resource languages faces the fundamental challenge of data scarcity. To address this limitation, researchers have developed multiple complementary approaches, ranging from rule-based error generation to statistical and neural methods. These approaches form a continuum of techniques that can be combined to create robust training datasets. Figure 3 provides an overview of methods.

Studies have used resources such as Wikipedia, CC100, the OSIAN, Lang-8 *corpus*, and e-books to generate synthetic corpora. They inject grammatical errors into correct sentences, creating pairs of original and error-laden sentences. This approach helps overcome the limitation of annotated corpora, improving GEC systems for low-resource languages.

Rule-based methods provide precise control over error types by applying linguistically informed transformations. This systematic approach intentionally introduces errors into grammatically correct sentences using a set of predefined linguistic rules. Specifically, it applies guidelines related to grammar, syntax, and punctuation to generate various error types, such as subject-verb agreement errors, incorrect tense usage, and misplaced modifiers. For example, *Sonawane et al. (2020)* applied this approach to Hindi, modifying the inflectional endings to create errors. Similarly, *Lee et al. (2021)* used phoneme differences and common errors to generate synthetic errors in Korean. For Basque, a study by *Beloki et al. (2020)* used a rule-based approach combined with ontology to create and assess syntactically accurate Basque sentences, where errors are introduced systematically based on linguistic rules.

Complementing rule-based approaches, pseudo-data generation creates learner-like corpora from monolingual sources. Pseudo-data generation involves injecting typical learner errors into sentences from monolingual datasets. For example, *Takahashi, Katsumata & Komachi (2020)*, created a learner *corpus* to train GEC models, improving performance. *Fang et al. (2023a)* also proposed using translationese, which shares stylistic characteristics with non-native texts, as an alternative data source for more robust GEC models. This approach adapts BERT-based classifiers to identify translationese effectively in parallel corpora and integrates errors to create more robust models for grammatical error correction.

More advanced noising techniques introduce controlled variations through operations such as swapping, deletion, insertion, and substitution (*Li et al., 2022*). Studies by *Solyman et al. (2021)* and *Cotet, Ruseti & Dascalu (2020)* have used these methods to generate large-scale synthetic data. Reverse noising, which applies noise during beam search, has been shown to enhance the performance of the model (*Kwon et al., 2023*). *Solyman et al. (2023)* proposed a broader noising framework incorporating strategies such as misspelling, swapping, token replacement, and reverse noising, which applies noise during beam search. Other noising techniques, as described by *Náplava et al. (2021)*, include altering

Synthetic Data Generation

**Rule-based**

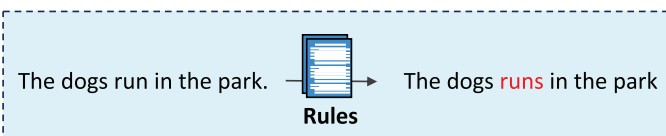

**Machine Translation**

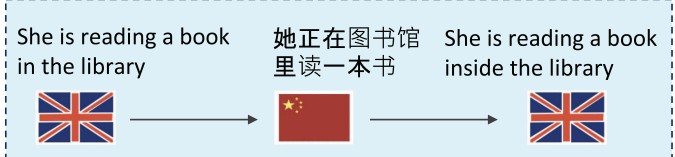

**Pseudo Data Generating**

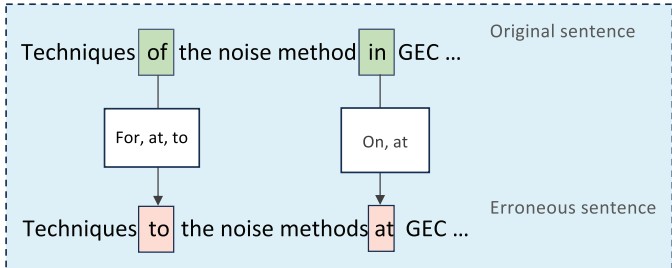

**Noising Methods**

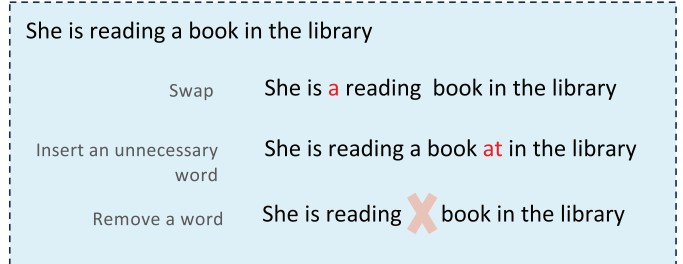

Pre-training Models

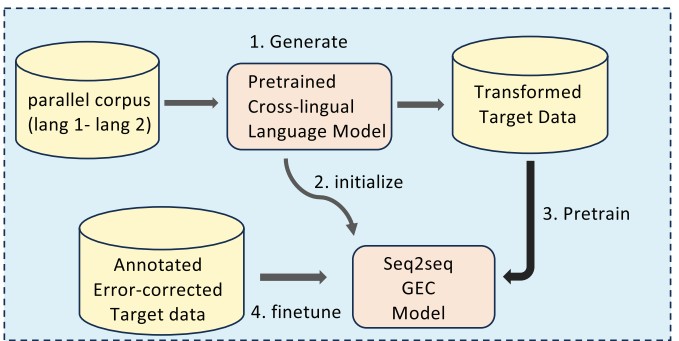

Cross-lingual and Multilingual Model

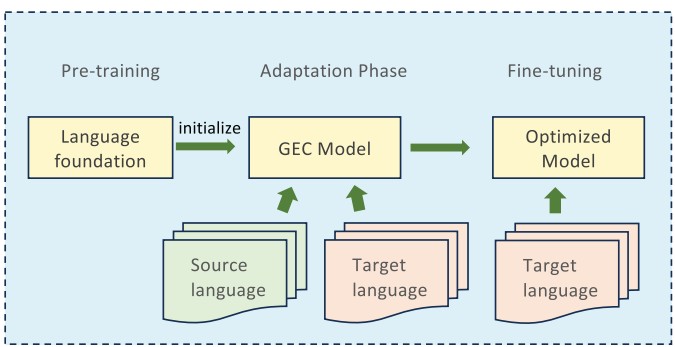

**Figure 3 An illustration of handling data scarcity techniques.**

diacritics, casing, suffixes, prefixes, and punctuation, among others. Furthermore, *Mahmoud et al. (2023)* proposed a semi-supervised method for confusion known as the equal distribution of synthetic error (EDSE) specifically for Arabic. This approach consists of two separate pipelines: the first pipeline creates errors including misspellings, punctuation mistakes, and structural errors in sentences (which involve swapping sentence components), while the second pipeline addresses more intricate errors related to syntax and semantics, employing various techniques such as tense unification through part of speech (POS) tagging, word removal, reordering, and substitution.

Machine translation approaches provide an additional dimension for data augmentation. Methods such as unidirectional translation and back-translation (BT) are valuable for generating parallel synthetic data. Back-translation involves translating text from the original language to another language and then goes back to the original, while unidirectional translation refers to translating text from one language to another in a single direction. Both methods help generate larger and more varied training data while

**Table 1 Impact of synthetic data generation on GEC performance across various languages, models, and augmentation techniques.**

| Authors | Lang | Model | Augmentation | Metric | ↑ Score (%) |
|---|---|---|---|---|---|
| Davidson et al. (2020) | [SPA] | NMT | Train on Artificial noised Wikipedia data (insert, del, scrambling), and finetune on COWS-L2H | $F_{0.5}$ | 0.024 → 0.224 |
| Sonawane et al. (2020) | [HI] | Transformer | Artificial data by using POS tagging and change inflectional ending | $F_{0.5}$ | 0.31 → 0.49 |
| Solyman et al. (2022) | [AR] | Transformer | Backtranslation + noise injections | $F_1$ | 63.79 → 67.66 |
| Flachs, Stahlberg & Kumar (2021) | [RU] | Transformer | Aspell + Unimorph | $F_{0.5}$ | 32.91 → 35.95 |
| Flachs, Stahlberg & Kumar (2021) | [CE] | Transformer | Aspell + Unimorph | $F_{0.5}$ | 71.90 |
| Flachs, Stahlberg & Kumar (2021) | [GE] | Transformer | Train on Artificial → finetuned wikiedits | $F_{0.5}$ | 58.00 → 66.74 |
| Flachs, Stahlberg & Kumar (2021) | [SPA] | Transformer | Train on Artificial → finetuned wikiedits | $F_{0.5}$ | 47.35 → 52.56 |
| Palma Gomez, Rozovskaya & Roth (2023) | [UK] | MT5 Large | Spell-based transformation, POS-based transformation and Back-translation | $F_{0.5}$ | 57.83 → 65.45 |
| Ingólfsdóttir et al. (2023) | [ICE] | ByT5 | Rule + noise | GLEU | 85.9 → 87.4 |
| Bondarenko et al. (2023) | [UK] | NMT | Punctuation, grammar, and round translation | $F_{0.5}$ | 0.601 → 0.620 |
| Solyman et al. (2023) | [AR] | Transformer | Misspelling (spell error, swap, reverse, replace) | BLEU | 73.20 → 75.22 |
| Mahmoud et al. (2023) | [AR] | Transformer | EDSE (1. spelling error, sentence structure, punctuation. 2. syntax and semantic error) compared with semi confusion method | $F_1$ | 45.06 → 50.48 |

preserving meaning. *Solyman et al. (2022)* combined back-translation with noising techniques to enhance GEC datasets. *Palma Gomez, Rozovskaya & Roth (2023)* proposed a novel approach that combined back-translation with spell-based transformations, producing multiple translation hypotheses that generate varied synthetic errors. This method translates sentences from Ukrainian into English, producing various leading translation hypotheses. These hypotheses are then translated back into Ukrainian to generate multiple backtranslations. In contrast to standard back-translation techniques that rely on entire back-translated sentences, this approach zeros in on particular confusion sets, generating synthetic errors that are closely related to the original sentences.

An additional promising strategy reconsiders the idea of employing round-trip machine translation (RTMT) for GEC (*Kementchedjhieva & Søgaard, 2023*). The researchers investigate the possibilities of RTMT in five different languages, using modern neural machine translation (NMT) models that have demonstrated significant advancements. In summary, as illustrated in Table 1, the impact of synthetic data on the GEC model improves the accuracy of the model.

Pre-trained models represent a valuable approach to generating synthetic data. For instance, *Sun et al. (2022)* used a cross-lingual model (PXLM) with non-autoregressive translation (NAT) to generate rough translations with grammatical errors, followed by post-editing to simulate common errors. Recent progress, as discussed by *Do, Nguyen & Nguyen (2025)*, utilizes zero-shot multilingual semantic parsing complemented by LLM-driven data augmentation. This approach creates precise semantic representations in

target languages, producing augmented data without the need for examples in the target languages, resulting in significant performance enhancements even with scarce data. These methods emphasize the capability of pre-trained models to tackle data shortages and boost GEC performance in low-resource, multilingual contexts.

The comparative effectiveness of these data generation strategies varies by language and available resources. Table 1 summarizes their impact across various languages, revealing that hybrid approaches combine multiple generation techniques consistently outperform single-method approaches.

## Knowledge transfer and model adaptation approaches

In addition to data augmentation techniques, knowledge transfer and model adaptation strategies offer promising solutions for low-resource GEC. These approaches leverage existing linguistic knowledge and pre-trained models to address the scarcity of annotated data.

Cross-lingual transfer learning has emerged as an effective approach for languages with limited annotated data. By transferring grammatical knowledge from high-resource languages, *Yamashita et al. (2020)* demonstrated that the performance of GEC can improve for low-resource languages. Multilingual models such as mBART-25 (*Liu et al., 2020*), fine-tuned for GEC, have also shown success in multiple languages, although balancing diverse grammatical structures remains a challenge (*Pajak & Pajak, 2022*).

The study by *Luhtaru, Korotkova & Fishel (2024)* demonstrated that the pre-trained multilingual machine translation (MT) models excel in GEC for low-resource languages when fine-tuned with error correction data. Their approach outperformed similarly sized models, proving particularly efficient in multilingual contexts. This finding underscores the potential for adapting translation-focused architectures for grammatical error correction tasks.

Pre-training and fine-tuning techniques have become central to addressing data scarcity. *Rothe et al. (2021)* proposed generating synthetic data through language-agnostic methods, followed by fine-tuning on language-specific datasets. This approach has shown encouraging results in various languages and offers a promising method for low-resource languages. The effectiveness of this two-stage process lies in its ability to learn general patterns of grammatical errors before adapting to language-specific characteristics.

The combination of synthetic pre-training with targeted fine-tuning has proven especially effective. Studies have shown that even modest amounts of authentic annotated data can significantly enhance model performance when built upon pre-trained foundations. *Flachs, Stahlberg & Kumar (2021)* established that models pre-trained on synthetic data and then fine-tuned on just 15,000 sentences of gold standard corrections can achieve impressive performance gains, emphasizing quality over quantity in the fine-tuning process.

## Summary and future directions

Combining synthetic data generation, pre-training and fine-tuning on curated datasets, and multilingual approaches provides a robust framework for mitigate data scarcity in

low-resource GEC. Synthetic methods such as noise injection, adversarial generation, and back-translation, when integrated with pre-training on pseudo-error data and the use of cross-lingual resources, ensure that GEC systems are both scalable and adaptable for diverse linguistic contexts.

The most promising directions involve hybrid approaches that intelligently combine multiple techniques. For example, *Sun et al. (2022)* demonstrated that the integration of cross-lingual pre-training with targeted fine-tuning and synthetic data augmentation yields performance gains that exceed what any single approach can achieve. This synergistic combination addresses both the quantity limitations of low-resource datasets and the quality requirements for effective error correction.

Despite these advances, significant challenges remain. Current approaches require considerable adaptation for morphologically rich and typologically diverse languages. Methods that perform well for languages structurally similar to high-resource ones may falter when applied to agglutinative or polysynthetic languages. Future research should focus on developing more flexible architectures that can better accommodate diverse linguistic features without requiring extensive language-specific engineering.

## ADDRESSING LINGUISTIC DIVERSITY AND TYPOLOGY

### Linguistic challenges and adaptation strategies

Addressing linguistic diversity in GEC systems requires customized approaches that account for the specific typological characteristics of individual languages. The most significant challenges arise from the structural and grammatical diversity across language families, particularly when developing systems for low-resource contexts.

**Morphologically rich languages**, such as Turkish and Finnish, create complex challenges for GEC systems. These agglutinative languages systematically combine morphemes to form highly inflected or compound expressions, resulting in words that carry substantial grammatical information through affixation (*Flachs, Stahlberg & Kumar, 2021*). This extensive morphological variation creates data sparsity issues for token-based models, as a single word may have numerous valid forms depending on its grammatical context. For example, in Turkish, the word "evlerimdekilerdendi" combines the root "ev" (house) with multiple suffixes to mean "was one of those in my houses." This morphological complexity creates unique challenges for GEC systems.

- Data sparsity becomes more severe as the number of possible word forms increases exponentially, with each additional morpheme.
- Traditional token-based models struggle to recognize errors when a single morphological mistake can make an entire word unrecognizable.
- Correcting such errors requires understanding complex morphological rules that determine the valid combinations of affixes and their ordering.

**Free-word-order languages** such as Russian, Greek, and Hindi introduce additional complications because grammatical relationships are not limited to fixed-word positions in sentences (*Korre & Pavlopoulos, 2022*). For example, in modern Greek, "O Giannis

diavazei to vivlio" (John reads the book) remains grammatically correct when rearranged as "To vivlio diavazei o Giannis" (The book reads John, meaning "John reads the book") because the case markings, not word order, indicate the subject and object relationships. This syntactic flexibility creates significant challenges for GEC systems:

- Sequence-to-sequence models trained on languages with fixed word order struggle to identify errors in free-word-order contexts, often incorrectly flagging valid syntactic variations as errors.
- Reference-based evaluation metrics such as BLEU and ERRANT penalize legitimate alternative word arrangements, making evaluation problematic.
- Classifying error types becomes more complex because errors related to word order in English may be mere stylistic variations in these languages.

**Code-switching** presents another significant challenge, particularly in multilingual communities where speakers alternate between languages within a single conversation or sentence (*Potter & Yuan, 2024*). This phenomenon creates distinct grammatical structures that challenge monolingual GEC systems, especially in linguistically diverse regions such as South and Southeast Asia. The presence of multiple syntactic, semantic, and grammatical rules from different languages increase complexity, requiring systems to dynamically adapt to language changes and reconcile mismatches between training data and real-world usage. This is particularly challenging for low-resource languages, where multilingual communication is common but specialized resources for code-switched grammatical analysis are scarce.

To address these diverse linguistic challenges, researchers have developed language-specific adaptations focusing on morphology-aware processing and specialized tokenization strategies. For Icelandic, byte-level tokenization has proven superior to traditional subword tokenizers such as byte pair encoding (BPE) (*Ingólfsdóttir et al., 2023*). By encoding text at the byte level, this technique effectively handles Icelandic complex morphology and mitigates issues with out-of-vocabulary words. Similarly, for Arabic, contextual preprocessing techniques highlight and model complex grammatical features, with GED serving as an auxiliary input alongside GEC tasks to improve model performance (*Alhafni et al., 2023*).

Modern Greek's complex inflectional structure and free word order have necessitated the integration of linguistic rules into neural models (*Korre & Pavlopoulos, 2022*). Pre-trained multilingual text-to-text transformers, such as mT5, have been fine-tuned for Modern Greek, achieving competitive results by leveraging error-specific rule-based preprocessing and balancing subword tokenization with syntax-level context augmentation. These adaptations effectively address the unique linguistic characteristics of modern Greek, improving the performance of neural architectures on this typologically distinct language.

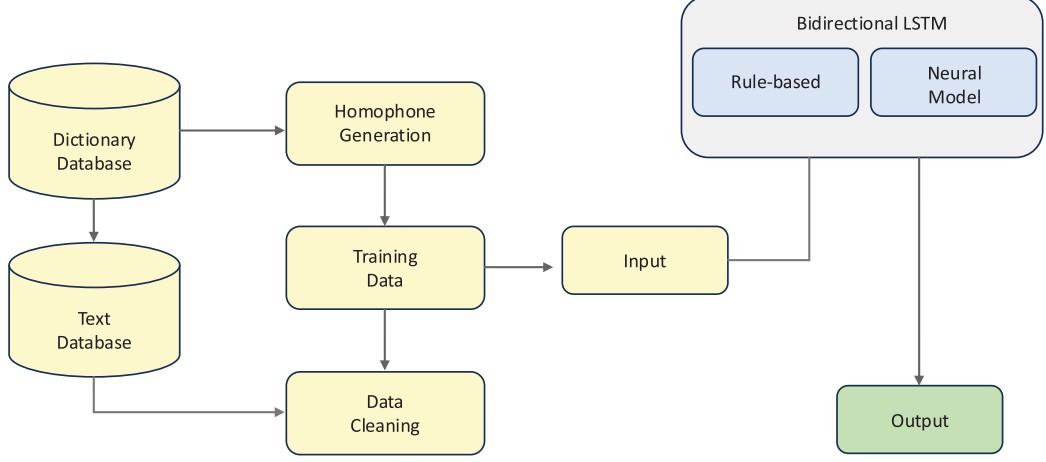

**Figure 4** An example of a hybrid rule-based and neural approach.

## Hybrid rule-based + neural approaches

A particularly promising direction for handling linguistic diversity in GEC systems is the hybrid approach, which combines rule-based linguistic knowledge with neural learning capabilities. This integration enables systems to address both specific grammatical rules and broader language patterns effectively, creating robust solutions for typologically diverse languages.

For non-native Arabic speakers, *Althafir & Ghnemat (2022)* proposed a hybrid model combining rule-based systems with AraBERT (*Antoun, Baly & Hajj, 2020*), a specialized version of BERT for Arabic. This approach leverages explicit linguistic knowledge about the rich morphological system of Arabic while benefiting from the contextual understanding capabilities of neural models. The rule-based component focuses on well-defined error patterns common among non-native speakers, while AraBERT contributes broader contextual understanding, particularly for semantic and pragmatic aspects of error correction.

Tibetan GEC presents unique challenges due to its distinct writing system and grammatical structure. *Jiacuo et al. (2020)* addressed these challenges by combining rule-based techniques with Bi-LSTM networks to target specific genitive and ergative errors common in Tibetan writing. This hybrid approach leverages linguistic expertise about Tibetan case marking while allowing the neural component to capture patterns that might be difficult to formalize in explicit rules.

For Bangla, *Noshin Jahan et al. (2021)* proposed a hybrid bidirectional long short-term memory (Bi-LSTM) and N-gram language model approach for GEC, as illustrated in Fig. 4. This system integrates dictionary and text databases with neural processing to handle the language's complex orthography and grammar, particularly focusing on real-word errors that maintain syntactic validity but violate semantic constraints. Their

evaluation showed that the hybrid approach achieved 82.86% precision in correcting real-word errors, significantly outperforming standalone neural models (59.16%).

Similar hybrid approaches have shown promise in other typologically diverse languages. *Jayasuriya et al. (2023)* implemented a three-stage hybrid method for Sinhala that integrated Google machine translation, rule-based strategies, and dependency parsing to correct verb agreement and object errors. This multistage approach addresses the agglutinative aspects of Sinhala while leveraging broader resources through machine translation components.

Punjabi GEC has been improved by combining rule-based techniques with machine learning methods, as demonstrated by *Verma & Sharma (2022)*, who integrated morphological analysis with part-of-speech tagging based on LSTM. This approach is particularly well-suited to Punjabi's complex morphology and syntactic patterns. Similarly, the North Sami GEC was improved by *Wiechetek et al. (2021)*, who employed a combination of rule-based and neural network techniques to address the rich morphological system of the language and relatively limited resources. They train a Bi-RNN based neural network. The precision of the rule-based model tested on a *corpus* with real errors (81.0%) is slightly better than the neural model (79.4%).

These hybrid models highlight the flexibility and effectiveness of combining linguistic rules with machine learning, offering robust solutions for low-resource languages and challenging linguistic typologies. The integration of explicit linguistic knowledge has proven especially valuable when handling languages with complex morphological systems or distinctive syntactic structures that might be underrepresented in training data.

## Summary and future directions

Addressing linguistic diversity in GEC systems requires customized approaches for each language family. Morphologically rich languages benefit from specialized tokenization, while code-switching requires dynamic multilingual models. Hybrid methods that combine rule-based knowledge with neural architectures show particular promise for low-resource settings. The most effective approaches adapt to specific typological features rather than applying models trained on high-resource languages without adaptation.

Future research should focus on extending these techniques across more language typologies while developing adaptive evaluation metrics for diverse grammatical systems. Priority should be given to models that generalize across related languages and employ meta-learning to adapt quickly to new linguistic patterns. Linguistically informed data augmentation techniques tailored to specific typologies could enhance performance while reducing annotation requirements.

In conclusion, while significant progress has been made in adapting GEC systems to linguistic diversity, continued integration of linguistic theory with computational approaches will be essential for creating truly inclusive grammatical error correction systems that can serve the world's rich tapestry of languages.

# COMPUTATIONAL TECHNIQUES AND MULTILINGUAL MODELS

## Neural architecture approaches

Neural sequence-to-sequence (seq2seq) models, particularly those utilizing the transformer architecture, have emerged as the dominant paradigm in GEC. By framing GEC as a translation task: converting ungrammatical sentences into their corrected forms, Transformer-based systems have surpassed traditional approaches for languages across diverse typological families.

Building on this foundation, neural machine translation has been widely applied across diverse linguistic contexts (*Ranathunga et al., 2023*), highlighting its adaptability in various linguistic contexts such as Arabic (*Alhafni, Habash & Bouamor, 2020*), Russian (*Trinh & Rozovskaya, 2021*), Spanish (*Davidson et al., 2020*), Japanese (*Koyama et al., 2020*), Thai (*Lertpiya, Chalothorn & Chuangsuwanich, 2020*), Vietnamese (*Nguyen, Dang & Nguyen, 2020*), and Basque (*Beloki et al., 2020*). The SCUT AGEC is an example of a framework for Arabic GEC that combines CNN with attention mechanisms to capture contextual features and dependencies. This framework used synthetic data generation as a strategy to mitigate the challenges associated with the limited availability of parallel data in GEC.

**Transformer models** have demonstrated remarkable effectiveness for GEC tasks in multiple languages (*Vaswani et al., 2017*). For example, *Cotet, Ruseti & Dascalu (2020)* and *Kurfalı & Östling (2023)* applied Transformers to Romanian and Swedish GEC respectively, confirming their adaptability to different linguistic contexts. Arabic GEC has similarly benefited from customized Transformer implementations, with studies by *Solyman et al. (2022, 2023)* adjusting model parameters such as layer number and batch size to optimize performance for Arabic's morphological complexity.

To overcome limitations in unidirectional decoding, *Mahmoud et al. (2023)* introduced bidirectional decoding, employing left-to-right and right-to-left decoders with Kullback-Leibler divergence regularization. This innovation effectively reduces exposure bias and has shown superior results compared to standard Transformers in low-resource languages like Arabic. In addition, multitask settings that combine GEC with neural machine translation have proven to be effective for languages such as Czech and English by specifically targeting common error types (*Náplava et al., 2021*).

**Copy-augmented transformers** (*Zhao et al., 2019*) represent an significant advancement in transformer architecture for GEC, as illustrated in Fig. 5. This approach extends the standard Transformer by incorporating a copying mechanism that enables direct replication of tokens from the input sequence, which is particularly valuable for handling rare or unseen words. This architecture has shown notable improvements for Hindi (*Sonawane et al., 2020*), Korean (*Lee et al., 2021*), and Russian (*Takahashi, Katsumata & Komachi, 2020*). The copy mechanism combines two probability distributions: the standard generative distribution for vocabulary prediction and a copying distribution based on attention alignment scores. This proves especially valuable for preserving proper

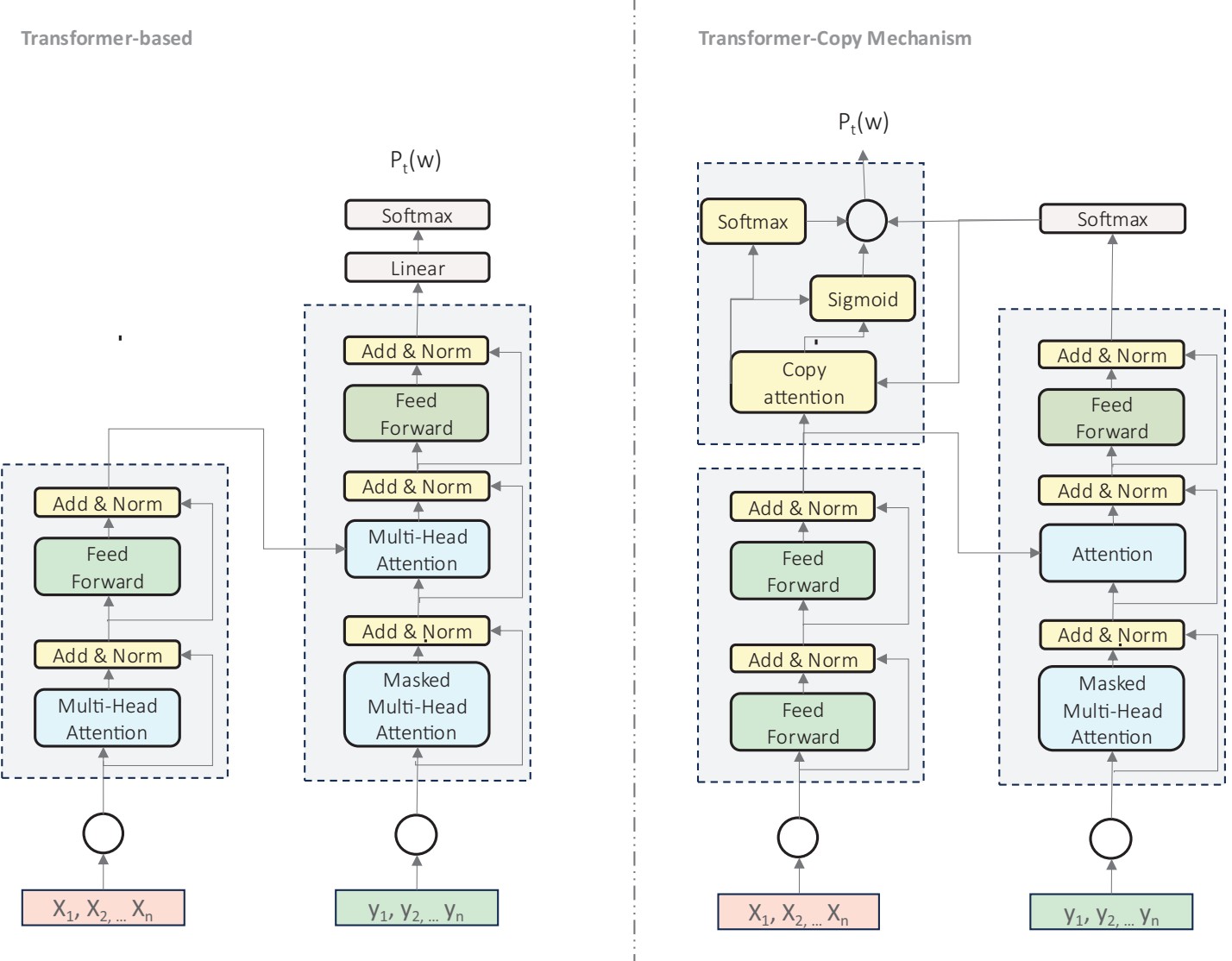

**Figure 5** An illustration of transformer-based model and modified transformer with copy mechanism.

names, handling spelling errors, and addressing out-of-vocabulary words, making it particularly suited to languages with rich morphology or limited resources.

Although transformer models offer powerful capabilities, edit-based approaches have gained popularity for their computational efficiency. Rather than generating entire corrected sentences, edit-based methods produce specific operations (replacements, insertions, deletions) that can be applied to erroneous text. This approach typically achieves inference speeds that are five to ten times faster than full-sentence generation, making it well suited for real-time applications in resource-constrained environments.

GECToR exemplifies this approach by iterative sequence tagging (*Omelianchuk et al., 2020*). For example, Balarila, a GEC model for Filipinos, adopts this methodology by training grammatically incorrect and correct sentences in a two-stage process. During inference, each token receives a transformation tag (such as REPLACE to substitute a token or ADD_PUNC to insert punctuation), with corrections applied iteratively until the sentence is fully corrected. This contextual, step-by-step approach ensures precise error resolution while maintaining computational efficiency.

RedPenNet (*Didenko & Sameliuk, 2023*) offers another edit-based solution through semi-autoregressive prediction of edit operations. This model achieves high accuracy while reducing computational complexity, as demonstrated by its competitive performance on benchmarks including BEA-2019 and UA-GEC+Fluency. The edit-based paradigm represents a valuable complement to transformer-based approaches, particularly in contexts where computational resources are limited.

Despite their strengths, neural architectures face challenges in low-resource settings. These models require substantial annotated data for optimal performance, which poses a significant constraint for low-resource languages. Data scarcity can lead to overfitting and difficulties in capturing complex linguistic features. Common mitigation strategies include synthetic data generation, noise injection, and unsupervised learning approaches that create pseudocorpora from monolingual data. While effective, these methods remain constrained by data quality and the suitability of pre-trained models for diverse linguistic typologies.

## Multilingual modeling strategies

Multilingual modeling has revolutionized GEC for low-resource languages by enabling the sharing of knowledge across language boundaries. These approaches leverage common linguistic patterns to overcome data limitations in specific languages while maintaining sensitivity to language-specific grammatical structures.

**Transformer-based multilingual** models like mT5 (multilingual text-to-text transformer) and PXLM (pre-trained cross-lingual language model) have demonstrated remarkable effectiveness in GEC by learning shared linguistic patterns across languages. These models, when fine-tuned for GEC tasks, facilitate the transfer of grammatical knowledge from resource-rich to low-resource languages.

The study by *Xue et al. (2021)* demonstrated that the fine-tuning of mT5 for GEC in various languages—including Greek (*Korre & Pavlopoulos, 2022*), Indian languages (*Ramaneedi & Pati, 2023*), Ukrainian (*Palma Gomez, Rozovskaya & Roth, 2023*), and other European languanges such as German, Russian and Czech (*Kementchedjhieva & Søgaard, 2023*)—significantly improves grammatical error detection and correction. Similarly, T5 models have been successfully fine-tuned for Russian GEC, in which the models assess response accuracy in grammar exercises (*Katinskaia & Yangarber, 2023*). PXLM has also been shown to be effective in generating synthetic data from source-target sentence pairs using non-autoregressive translation methods, achieving competitive results on benchmark for Chinese (NLPCC 2018) and Russian (RULEC-GEC) (*Sun et al., 2022*).

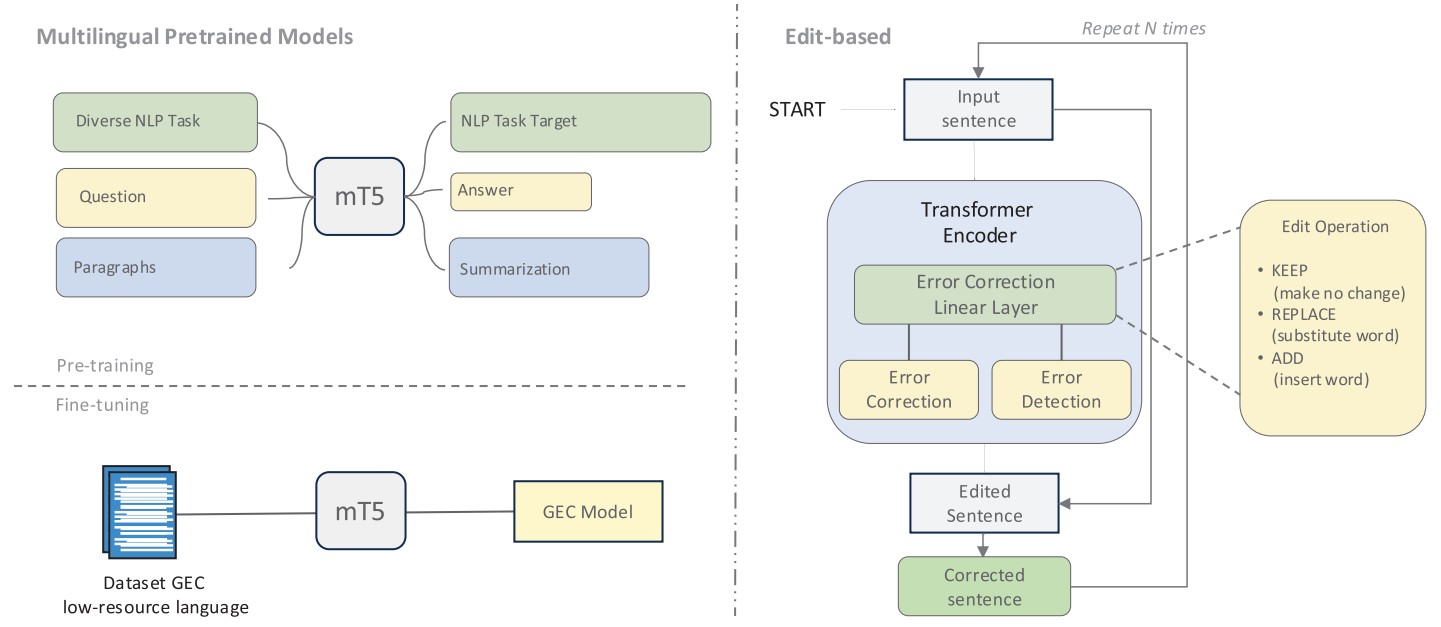

**Figure 6** An illustration of multilingual pretrained models and an edit-based approach.

Language-specific adaptations of pre-trained models have shown particular promise, as illustrated in Fig. 6. For Arabic, specialized models such as AraBART (*Kamal Eddine et al., 2022*) and AraT5 (*Nagoudi, Elmadany & Abdul-Mageed, 2022*) have been adapted for GEC tasks (*Alhafni et al., 2023*). Similarly, the T5 and ByT5 models have been fine-tuned for languages like Lithuanian (*Stankevičius & Lukoševičius, 2022*) and Icelandic (*Ingólfsdóttir et al., 2023*), with ByT5's byte-level tokenization proving especially effective at capturing language-specific nuances. This byte-level approach offers particular advantages for languages with complex morphological characteristics and spelling errors by focusing on fundamental byte-level representations rather than vocabulary-constrained tokens.

Other multilingual adaptations include mBART-50 (*Tang et al., 2021*), which has been fine-tuned for Ukrainian GEC, and multilingual RoBERTa (*Liu et al., 2019*), which has been adapted for tasks such as comma insertion in Czech (*Machura, Frémund & Švec, 2022*) and reposition handling in Russian L2 GEC (*Remnev et al., 2023*). For Thai, the WangchanBERTa model (*Lowphansirikul et al., 2021*) has been used to correct common misspellings, further demonstrating the effectiveness of pre-trained language models in low-resource contexts (*Pankam, Limkonchotiwat & Chuangsuwanich, 2023*).

## Advanced generative and large language model approaches

Advanced generative techniques and large language models (LLMs) represent the cutting edge of GEC research, offering promising solutions for languages with limited resources through few-shot learning and generative capabilities.

**Generative adversarial networks (GANs)** have emerged as powerful tools for generating synthetic data in GEC systems, particularly valuable in low-resource settings where

**Adversarial Approaches and GANs**

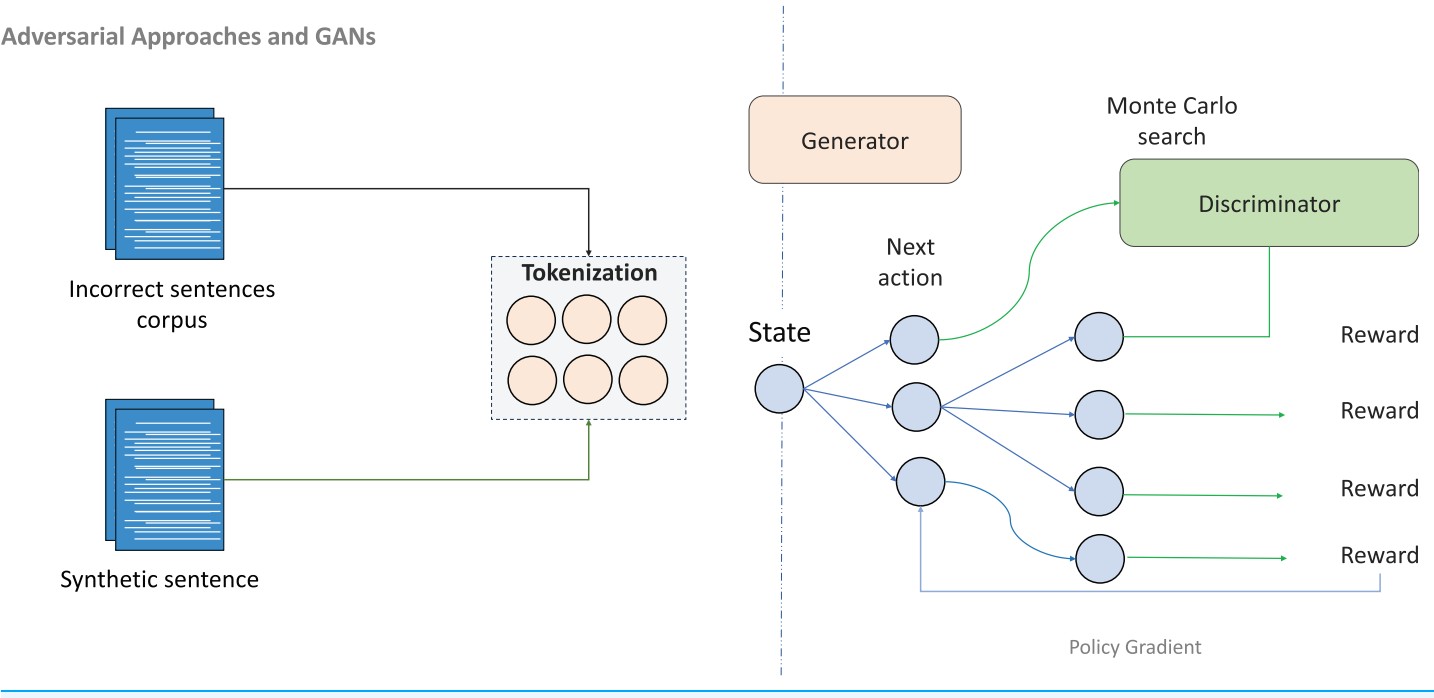

**Figure 7 An illustration of adversarial approaches and GANs.**

authentic error-corrected data are scarce. A modified sequence GAN was successfully applied to Persian GEC, generating errors aligned with the distinctive grammatical patterns of the language, including verb positioning and subject–verb agreement rules (*Golizadeh, Golizadeh & Forouzanfard, 2022*). By combining synthetic adversarial data with rule-based linguistic adaptations, this approach improved model robustness and F-score performance, demonstrating the potential of adversarial methods for languages lacking annotated corpora.

As illustrated in Fig. 7, adversarial training creates a competitive dynamic between generator and discriminator components, enabling the production of increasingly realistic grammatical errors. This approach proves especially valuable for simulating the diversity of learner errors in languages with limited annotated data, creating paired data that balance grammatical plausibility with error realism.

Recent research has increasingly focused on LLMs such as Llama and GPT for low-resource GEC tasks. These models demonstrate remarkable capabilities in few-shot and fine-tuning scenarios, producing high-quality corrections with minimal language-specific training data. ChatGPT-3.5 Turbo has been applied to Korean GEC (*Park et al., 2024*), Esperanto (*Liang, 2024*), and Brazilian Portuguese (*Penteado & Perez, 2023*), demonstrating its versatility in various linguistic structures. Additional studies have used ChatGPT for Arabic GEC (*Kwon et al., 2023*) and have comprehensively evaluated its performance in English and non-English contexts, including Chinese and German (*Fang et al., 2023b*).

# PeerJ Computer Science

**LLM-based**

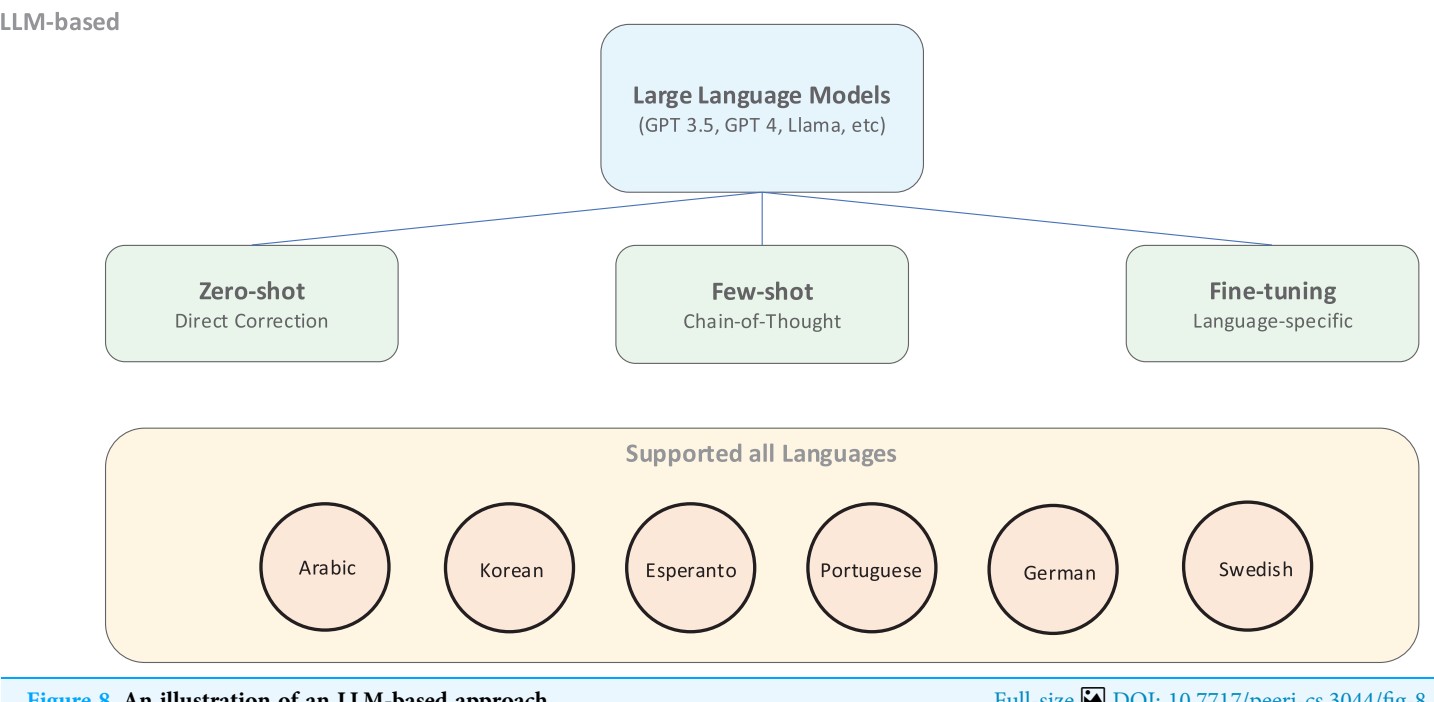

**Figure 8 An illustration of an LLM-based approach.**

As shown in Fig. 8, LLM-based approaches offer flexible deployment options ranging from zero-shot application to few-shot learning and fine-tuning for specific languages. These models particularly excel at zero-shot and few-shot chain-of-thought prompting strategies (*Kojima et al., 2023*), which prove especially valuable for low-resource and multilingual GEC where extensive training data are unavailable.

In an evaluation of GEC systems for Swedish learner texts, *Östling et al. (2024)* compared rule-based and LM-based systems, finding that GPT-3 significantly outperformed conventional systems in grammatical accuracy and fluency, although it was less effective at preserving meaning compared to human corrections. This research suggests that current evaluation methods may favor grammatical precision over semantic integrity, highlighting the need for human post-editing to achieve balanced GEC performance assessment.

Beyond multilingual models, language-specific LLMs have been developed to address unique linguistic challenges. RoGPT2 for Romanian achieved state-of-the-art results in the RONACC *corpus* (*Niculescu, Ruseti & Dascalu, 2021*), while unigram, bigram, and trigram word-level modeling has been successfully applied to Tamil GEC (*Sakuntharaj & Mahesan, 2021*), demonstrating the adaptability of language-specific models for various grammatical contexts.

Although modern neural approaches dominate current research, statistical machine translation (SMT) and classification-based methods continue to offer valuable approaches for certain low-resource contexts. SMT applies probabilistic models to predict word alignments from parallel data, as demonstrated in a Punjabi GEC system

(*Jindal, Singh & Sharma, 2021*). Similarly, the multiple filter correction framework (MFCF) for Indonesian GEC (*Zheng, Lin & Shengyi, 2022*) employs dictionary filtering and scoring models to enhance accuracy in nonword error correction. The MFCF framework integrates dictionary filtering, denoising of candidate words, and scoring models to rectify errors involving nonwords. By using the Window Search Algorithm (WSA) to minimize the search space and using a word vector-based scoring model (CWSM-WV) to evaluate potential words, this technique effectively limits the number of possible corrections, demonstrating the effectiveness of probabilistic filtering methods in low-resource grammatical error correction (GEC).

Classification approaches focus on categorizing grammatical errors rather than generating corrections directly. The Arabic GEC framework by *Alkhatib, Monem & Shaalan (2020)* used a Bi-LSTM model for error classification, while (*Lin et al., 2023*) framed Tagalog GEC as a multiclassification task using BERT. Although less common than generative methods, classification approaches offer effective, targeted error detection, offering valuable pathways to develop GEC systems in underrepresented languages.

The previous discussion has explored multiple computational methodologies applicable to GEC tasks in languages that exhibit typological variation. To provide a concrete comparison of these techniques, Table 2 presents empirical performance data for representative GEC systems across multiple languages and architectural paradigms.

## Summary and future directions

Computational techniques and multilingual models have significantly advanced GEC capabilities, particularly for low-resource languages. Transformer-based architectures remain central to state-of-the-art GEC systems, with innovations such as copy mechanisms and edit-based approaches that enhance their efficiency and accuracy. Multilingual pre-trained models have proven invaluable for knowledge sharing across languages, enabling effective transfer from high-resource to low-resource contexts. Meanwhile, adversarial approaches and large language models represent promising directions for addressing data scarcity through synthetic data generation and few-shot learning.

Future research should focus on improving multilingual pretraining techniques to minimize interference between languages with different typological characteristics. Improving the adaptability of LLMs for morphologically rich and syntactically diverse languages represents another crucial direction, potentially through specialized fine-tuning approaches that preserve linguistic diversity. The integration of adversarial methods for dynamic error simulation could further enhance data augmentation strategies, particularly for languages lacking authentic learner corpora.

As these technologies continue to evolve, the field is moving toward more scalable generalized GEC systems capable of addressing both resource scarcity and linguistic diversity in multilingual settings. These innovations will likely contribute to more inclusive and effective grammatical error correction tools that can serve the global linguistic landscape.

**Table 2 Comparative GEC approaches across languages show model performance by the architecture type.**

| Lang | Study | Model architecture | Dataset | Performance |
|---|---|---|---|---|
| [AR] | *Solyman et al. (2022)* | Transformer-based Bidirectional decoding with KL divergence | QALB-2014 QALB-2015 | $F_1$: 71.82%, $F_1$: 74.18% |
| [AR] | *Mahmoud et al. (2023)* | Transformer-based Bidirectional Knowledge Distillation | QALB-2014 QALB-2015 | $F_1$: 71.51% $F_1$: 74.03% |
| [ID] | *Musyafa et al. (2022)* | Transformer-Copy | Custom Indonesian *corpus* | $F_1$: 71% BLEU: 78.13% |
| [ID] | *Lin et al. (2023)* | BERT-Based | Custom Indonesian *corpus* | $F_{0.5macro}$: 70.15% |
| [ES] [UK] [GE] | *Luhtaru et al. (2024)* | LLM-Based Llama | Estonian Ukrainian German | $F_{0.5}$: 69.97%, $F_{0.5}$: 74.09%, $F_{0.5}$: 76.75% |
| [UK] | *Palma Gomez, Rozovskaya & Roth (2023)* | Multilingual pre-trained model mT5-Large + synthetic data | UA-GEC + Fluency | $F_{0.5}$: 65.45% |
| [UK] | *Didenko & Sameliuk (2023)* | Edit-Based RedPenNet | UA-GEC + Fluency | $F_{0.5}$: 67.71% |
| [KO] | *Park et al. (2024)* | LLM-Based ChatGPT-3.5 zero-shot | K-NCT | BLEU: 46.58% |
| [KO] | *Lee et al. (2021)* | Transformer-Copy | Custom Korean *corpus* | $F_{0.5}$: 74.86% |
| [IS] | *Ingólfsdóttir et al. (2023)* | Multilingual pre-trained model ByT5 | IceEC + Synth 550M | GLEU: 87.4% |
| [ES] | *Davidson et al. (2020)* | NMT bi-directional LSTM encoder and an LSTM decoder | COWS-L2H | $F_{0.5}$: 22.4% |
| [HI] | *Sonawane et al. (2020)* | Transformer-Copy | Custom Hindi *corpus* | $F_{0.5}$: 49% GLEU: 80% |
| [BN] | *Hossain et al. (2024)* | Transformer-Based Panini model | BGEC | Accuracy: 83.3%, $F_1$: 0.833 |
| [CZ] | *Náplava et al. (2022)* | Transformer-Based | GECCC | $M_{0.5}^2$−score: 72.96% |
| [GR] | *Korre & Pavlopoulos (2022)* | Multilingual pre-trained model mT5 | GNC | $F_{0.5}$: 52.63% |
| [RO] | *Niculescu, Ruseti & Dascalu (2021)* | LLM-Based RoGPT2 | RONACC | $F_{0.5}$: 69.01% |
| [SW] | *Östling et al. (2024)* | LLM-Based GPT-3.5 | Custom Swedish *corpus* | GLEU: 63% |

# EVALUATION TECHNIQUES AND METRICS

## Evaluation challenges and metric adaptations

Evaluation of GEC systems poses unique challenges, particularly for low-resource languages, where linguistic diversity, typological complexity and limited annotated corpora introduce significant complications. Standard evaluation metrics developed for high-resource languages often fail to capture the nuances of morphologically rich or syntactically flexible languages, necessitating adaptations for effective assessment.

**BLEU**, originally designed for machine translation evaluation, calculates the overlap n-gram between the system output and the reference text. Although widely used, its reliance on surface forms makes it inadequate for morphologically rich languages, where valid corrections may exhibit substantial surface variation while maintaining grammatical correctness. This limitation becomes particularly pronounced in agglutinative languages, where a single word can express complex grammatical relationships through multiple morphemes.

**GLEU**, adapted specifically for GEC, offers improvements by scoring corrections by comparing the system output to gold standard references, measuring both precision and recall of n-grams. Although more robust than BLEU for English-focused tasks, GLEU still struggles with complex grammatical typologies, where word order flexibility and morphological variation create multiple valid corrections that may differ significantly at the surface level.

**ERRANT** provides a GEC-specific framework that evaluates edits made by a system relative to reference corrections. By offering fine-grained feedback on error types (*e.g.*, substitutions, insertions, deletions), ERRANT enables detailed analysis of system performance. However, this framework faces significant challenges in capturing language-specific freedoms in word order and morphology, particularly in languages where grammatical relationships are expressed through case marking rather than position.

Most existing GEC datasets and annotation tools are designed for high-resource languages, leaving low-resource languages with minimal resources for training and evaluation. Researchers have attempted to address this gap by identifying standard error categories and creating automatic annotation systems. However, structural variations among languages make it difficult to establish unified error categories that effectively span multiple linguistic contexts.

To address these challenges, researchers have developed language-specific adaptations of existing metrics and annotation systems. **ERRANT-TR** for Turkish has been tailored to accommodate the agglutinative structure and the rich morphology language (*Uz & Eryiğit, 2023*). This system first employs a modified Damerau-Levenshtein algorithm enhanced with linguistic features such as POS tags and lemmas to align erroneous tokens with their corrections. Errors are then categorized using rule-based labels ("R" for Replacement, "M" for Missing, or "U" for unnecessary). To accurately capture Turkish morphology, ERRANT-TR aligns morphemes and accounts for features such as tenseness and mood using a similarity threshold of 0.85. When tested in 106 manually annotated sentences, this system achieved a span-based $F_{0.5}$ score of 77.04%.

**KAGAS** for Korean annotates the error types by aligning erroneous and corrected sentences using morpheme-level POS tags and a modified Damerau-Levenshtein algorithm (*Yoon et al., 2023*). This tool defines fourteen Korean-specific error types, including spacing, particle use, and verb form errors, specifically designed to handle common Korean orthographical and syntactical issues. KAGAS also generates standardized M2 files, facilitating a consistent evaluation of Korean GEC models and establishing benchmarks for performance evaluation.

Similarly, **ARETA** for Arabic aligns erroneous and corrected phrases using a character-based edit distance algorithm before applying a series of specialized modules to identify specific error types (*Belkebir & Habash, 2021*). This system categorizes errors into punctuation, orthographic, morphological, syntactic, and semantic groups, using tools such as CAMeL for morphological analysis. ARETA has shown high effectiveness with a micro-average F1-score of 85.8%.

**ELERRANT** for Greek introduces an adapted version of ERRANT specifically designed for the complex morphology and grammar of modern Greek (*Korre, Chatzipanagiotou & Pavlopoulos, 2021*). This system employs the Greek Hunspell dictionary for spelling error detection and Greek SpaCy as the primary POS tagger. ELERRANT adjusts ERRANT's original categories to incorporate Greek-specific error types, consolidating some categories like noun inflection and possessive forms into broader categories such as NOUN to capture the language's inflectional nature better. Additionally, ELERRANT introduces two new error types: ACC for accentuation errors and FN for final numerical errors, both prevalent in Greek grammar.

For French, **FRETA-D** automates the annotation of grammatical and phonetic errors in dictations (*Luo, Zhai & Qin, 2022*). Using French-specific resources and rule-based algorithms, this toolkit preprocesses transcriptions, identifies errors at both the grammatical and phonetic levels, and categorizes them into syntactic, orthographic, and phonetic types. It effectively handles complex issues such as agreement, conjugation, and phonetic ambiguities, providing valuable support for language learning and assessment.

**RuERRANT** for Russian adapts the ERRANT framework to address the challenges of the morphologically rich structure of Russian (*Katinskaia et al., 2022*; *Rozovskaya, 2022*). This grammatical error annotation toolkit automatically extracts edit operations from parallel original and corrected sentences and classifies them by error type. To accommodate Russian's complex morphology, RuERRANT implements more fine-grained tags than its English counterpart, adding specialized categories for case, gender, aspect, mood, voice, comparative, and short/full forms of adjectives (*Katinskaia et al., 2022*).

The system employs a rule-based classification approach that preprocesses tokens with a morphological analyzer to produce base forms and generate all possible morphological analyses (*Rozovskaya, 2022*). This captures the intricate categories of Russian grammar including gender, number, declension, aspect, voice, person, and tense. The classification rules follow a detailed algorithm that distinguishes between morphological and lexical errors by analyzing POS tags, base forms, and linguistic categories, with special handling for verb aspect errors using verb pair lists and voice errors through specific ending patterns. Manual evaluation demonstrates that the resulting error classifications are considered correct or acceptable in 93.5% of cases (*Rozovskaya, 2022*). This adaptation enables the toolkit to handle the complex grammatical features of Russian, providing valuable data to study patterns of grammatical errors and to develop new exercises for language learners.

As discussed in the challenges section, these specialized metrics help address the fundamental limitations of standard evaluation approaches when applied to morphologically rich languages or languages with flexible word order. By incorporating language-specific features, these adaptations provide more accurate assessments of GEC system performance across diverse linguistic contexts. Table 3 presents a detailed comparison of the architectural design and implementation details of these specialized annotation tools in different languages.

**Table 3 Comparison of grammatical error annotation tools for different languages.**

| Tool | Lang. | Base | Key methods | Resources used |
|---|---|---|---|---|
| ERRANT-TR | [TR] | ERRANT | Rule-based classification; Modified Damerau-Levenshtein algorithm; Morpheme-level alignment; Hierarchical error structure | Morphological analyzer; POS tagger; Surface form similarity threshold (0.85) |
| KAGAS | [KO] | ERRANT | Modified Damerau-Levenshtein algorithm; Korean-specific linguistic costs; Custom merging rules; $M^2$ file format generation | Morpheme-level POS tags; Korean-specific linguistic costs; 14 Korean-specific error types |
| ARETA | [AR] | ERRANT | Levenshtein edit distance; Unsupervised approach; Three-part approach: alignment, annotation, and evaluation | CAMeL Tools; CALIMA-Star morphological analyzer; ALC Error Tagset (modified); 7 classes and 26 error tags |
| ELERRANT | [EL] | ERRANT | Rule-based error type classifier; Unsupervised approach; Error type merging | Greek Hunspell spellchecker; SpaCy POS tagger; Greek-specific error categories |
| FRETA-D | [FR] | N/A | Data-set-independent classifier; Error boundary identification; Fine-grained error classification | Framework with 25 main error types; French grammar rules; FFL learners' common error patterns |
| RuERRANT | [RU] | ERRANT | Rule-based classification framework; Edit operation extraction; POS and morphological classification | Morphological analyzer; Fine-grained tags for Russian morphology; Verb pair lists for aspect errors |

## Innovations in evaluation methodologies

To address the limitations of standard evaluation approaches, researchers have developed innovative methodologies that better account for linguistic diversity and multiple valid corrections for a single error.

The research by *Rozovskaya & Roth (2021)* highlights a critical disconnect between the standard evaluations of GEC systems and their actual performance. Examining the challenges faced in GEC evaluation, particularly regarding the accuracy and consistency of existing assessment techniques, the authors propose the "closest gold" standard to allow for more precise evaluations. This novel method demonstrates that GEC systems perform significantly better than traditional metrics suggest, particularly for lower-ranked hypotheses that provide valid, diverse corrections often overlooked by fixed-reference evaluations. This approach offers a comprehensive and accurate evaluation methodology that better captures the full range of acceptable grammatical corrections.

Building on this work, *Palma Gomez & Rozovskaya (2024)* enhance the evaluation of Russian GEC systems by introducing multi-reference benchmarks. Their approach incorporates two additional references (RU-Lang8 and RULEC) to supplement existing single-reference datasets, resulting in a more comprehensive and realistic evaluation framework. By accounting for multiple valid corrections, this methodology reduces the penalty for linguistically sound alternatives that differ from a single reference, offering a more accurate assessment of system performance.

**Reference-less evaluation** techniques represent another promising direction, bypassing the need for gold-standard references entirely. *Suzuki et al. (2022)* established a quality estimation dataset for Japanese GEC which demonstrated a stronger correlation with manual evaluations than traditional reference-based approaches. This method assesses grammatical correctness based on the qualities of the corrected text itself rather than its

similarity to predefined references, potentially offering more flexibility for languages with limited annotated resources.

These innovations address a fundamental challenge in GEC evaluation: the existence of multiple valid corrections for many grammatical errors. Traditional single-reference evaluation methods often penalize systems for producing grammatically correct alternatives that differ from the reference, failing to recognize the diversity of acceptable corrections. Multi-reference and reference-less approaches mitigate this limitation, providing more nuanced and fair assessments of GEC system performance.

For low-resource languages, these methodologies offer particular promise by reducing dependency on extensive gold-standard references. Quality estimation models can be trained on smaller datasets and potentially transferred across related languages, while multi-reference approaches can leverage diverse sources of corrections to construct more comprehensive evaluation frameworks. These innovations represent significant steps toward more accurate and linguistically informed evaluation methods that better accommodate the realities of grammatical error correction in diverse languages.

### Summary and future directions

Evaluation techniques for GEC systems require continued innovation, particularly for low-resource language. Although language-specific adaptations of standard metrics have shown promise, future research should focus on developing unified multilingual frameworks that remain sensitive to typological diversity. Innovations such as multi-reference evaluation and quality estimation models offer promising directions by reducing dependency on extensive gold-standard references and acknowledging multiple valid corrections.

The field should prioritize evaluation approaches that balance grammatical accuracy with semantic preservation, an especially important consideration for morphologically rich languages, while enabling fair cross-linguistic comparisons. These improvements will support the development of more inclusive and effective GEC systems for the world's diverse languages.

## DATASETS

### GEC *Corpus* in low-resource language

Constructing datasets for GEC research and evaluation presents multiple challenges, particularly when the goal is to produce large, high-quality datasets for languages with limited resources. The study by *Pajak & Pajak (2022)* demonstrates the critical role of both dataset size and quality in GEC performance. For example, although the Chinese language dataset is extensive, its reliance on community-generated annotations has led to lower accuracy, demonstrating the importance of high-quality annotations. Similarly, the limited Russian dataset, with only 5,000 tests, restricts the model's effectiveness.

To address these challenges, researchers have developed several specialized corpora for GEC in low-resource languages, as shown in Tables 4 and 5. These corpora provide essential resources that support error correction tasks by providing foundational data for

**Table 4 GEC corpora—language, *corpus* size, error rate and domain (part 1 of 2).**

| Lang | *Corpus* | Size | Err. rate | Domains |
|---|---|---|---|---|
| [AR] | QALB-2014 | 1M+ words | 14.1% | News comments |
| [AR] | QALB-2015 | 1M+ words (native) 91k words (L2) | 14.1% (native) 6.4% (L2) | News; L2 essays |
| [AR]/[EN] | ZAEBUC | 602 texts (214 AR, 388 EN) | AR:21.4% EN:11.8% | University essays |
| [BN] | BGEC | 7.07M sent. | N/A | Synthetic data from paraphrase *corpus* |
| [BO] | Tibetan GEC | 1.79M | 31.85% | Multi-domain |
| [CZ] | AKCES-GEC | 47k sent. | 21.4% | L2 essays; Romani essays |
| [CZ] | GECCC | 83k sent. | 5.8–26.2% | Native web; Romani essays; L2 essays |
| [EO] | Eo-GEC | 307 sent. | N/A | Textbook, user writing, exam resources |
| [ES] | COWS-L2H | 33,950 sent. | N/A | Essays; academic writing |
| [GR] | GNC | 227 sent. | N/A | Native high-school essays |
| [ID] | SPECIL | 21,500 sent. | N/A | e-books (education) |
| [IS] | IceEC | 4,044 texts | N/A | Student essays; online news; Wikipedia |
| [JA] | NAIST JSL | 2,042 sent. | N/A | Blogs; diary entries |
| [KO] | K-NCT | 3,000 sent. | N/A | Written, spoken, dialogue styles |
| [RO] | RONACC | 10,119 sent. | 65.1% POS; 14.5% spell; 13.5% other | TV/radio phrases; written mistakes |
| [RU] | RULEC-GEC | 12,480 sent. | 6.3% | University essays |
| [RU] | ReLCo | 15.5k sent. | 0.64–0.75% | Exercise items in learning system |
| [RU] | RU-Lang8 | 48,260 sent. | 15.6% (dev) 11.3% (test) | Online language-learning platform |
| [TH] | Thai UGWC | 15,576 sent. | N/A | Social media; web forums |
| [UK] | UA-GEC | 33,735 sent. | 8.2% | Essays; social media; chats; formal texts |

**Table 5 GEC corpora—languages, writer profiles, annotation schemes, and error types (Part 2 of 2).**

| Lang | *Corpus* | Writers | Annotation | Error types |
|---|---|---|---|---|
| [AR] | QALB-2014 | Native writers | 7 categories; M2 format | ORTH, GRAM, PUNCT |
| [AR] | QALB-2015 | Native and non-native learners | 7 categories; M2 format | ORTH, GRAM, PUNCT |
| [AR]/[EN] | ZAEBUC | 1st-year university students (90% female) | SPL, MORPH, POS | ORTH, GRAM, LEX |
| [BN] | BGEC | Synthetic | 10 categories | VERB:INF, AGR, MISS, homonyms, PUNCT |
| [BO] | Tibetan GEC | — | Rule-based pre-processing | Genitive, ergative case errors |
| [CZ] | AKCES-GEC | Non-native learners; Romani ethnolect | Error categories; M2 format | SPL, GRAM, PUNCT, DIAC |
| [CZ] | GECCC | Native students; web users; Romani; L2 learners | Czech-specific ERRANT; M2 format | 10 main error types + subtypes |
| [EO] | Eo-GEC | Esperanto learners; B2-level speakers | Fine-grained linguistic annotations | Replacement, MISS; NOUN:CASE; VERB:TENSE |
| [ES] | COWS-L2H | University students; L2 learners | Error annotations; parallel corrections | Spanish learner GEC; error detection assessment |
| [GR] | GNC | High-school students | Four-field annotation; error types + corrections | SPL, accent, PRON, PREP |

| Lang | *Corpus* | Writers | Annotation | Error types |
|------|----------|---------|------------|-------------|
| [ID] | SPECIL | Native writers | Six error types; 843-word real-word dictionary | Non-word (ins, del, sub, transposition); real-word; PUNCT |
| [IS] | IceEC | High-school students; web sources | 253 error codes; 31 sub category | PUNCT, WO, SPACE, non-word |
| [JA] | NAIST JSL | Japanese-as-second-language learners | Multiple references; minimal edits | Particle, style, TYPO |
| [KO] | K-NCT | Native university students | Error spans with locations; four main types | SPACE, PUNCT, numerical, SPL/GRAM (23 subcats) |
| [RO] | RONACC | Native speakers | Manual; ERRANT adaptation | POS, MORPH, ORTH, WO |
| [RU] | RULEC-GEC | 15 foreign + 13 heritage speakers | Manual; 23 error categories | SPL, NOUN:CASE, LEX |
| [RU] | ReLCo | L2 learners | Automatic + manual verification | GRAM, SPL, VERB:FORM, AGR |
| [RU] | RU-Lang8 | Diverse L1s | Automatic + manual re-annotation | rpl, ins, del |
| [TH] | Thai UGWC | Native Thai speakers | Six error types; manual annotation | Misspelling, morphed words, abbreviation, spoonerism, slang, other |
| [UK] | UA-GEC | Native and non-native speakers | Error-type annotations; GEC +Fluency variants | GRAM, fluency, SPL, PUNCT |

training GEC systems in underrepresented languages. By offering language-specific data, these resources address the scarcity of quality datasets and contribute to advancing GEC capabilities in low-resource contexts.

**AKCES-GEC** provides the first substantial Czech grammatical error correction *corpus*, composed of several subcorpora (*Náplava & Straka, 2019*). It includes CzeSL (Czech as a Second Language) featuring non-native speakers, ROMi, which represents the Romani ethnolect used by children and teenagers, and SKRIPT and SCHOLA, which contain texts written by native Czech students. The *corpus* contains token-level annotations with error types in M2 format, dividing the data into two primary domains with relatively high error rates (average 21.4%). This resource establishes a foundation for Czech GEC systems through its comprehensive error typology and diverse writer demographics.

**QALB-2014** introduced the first shared task on Arabic text correction, featuring comments from the Aljazeera news website by native Arabic speakers (*Mohit et al., 2014*). Containing 1.1 million words in training data with 306K annotated errors across seven action types, it highlights Edit operations (55.34%) and Add operations (32.36%) as most frequent. Although significant for Arabic GEC development, its limitation to online comments from a single source restricts broader application across different Arabic-speaking contexts and writing domains.

**QALB-2015** builds on QALB-2014 by incorporating the Arabic L2 learner texts in addition to native speaker content (*Rozovskaya et al., 2015*). Native texts come from Al Jazeera comments, while L2 texts derive from the Arabic Learner Written *Corpus* and the Arabic Learner *Corpus*. The error distributions show notable differences between the

groups, with L2 texts containing more Edit corrections (60.7% *vs.* 51.9%), reflecting acquisition challenges. Despite providing valuable insights into Arabic learner errors, the *corpus* is limited by its binary native/non-native categorization, lacking proficiency gradation.

**ZAEBUC** presents a unique bilingual *corpus* with texts by the same writers in Arabic and English (*Habash & Palfreyman, 2022*). It contains 388 English essays (87.6K words) and 214 Arabic essays (33.3K words) from 397 first-year UAE university students. The *corpus* features comprehensive annotations, including spelling corrections, morphological tokenization, POS tagging, and CEFR proficiency ratings for both languages. Its consistent cross-linguistic annotation standards enable direct comparability, though limitations include modest size, limited topic range, and gender imbalance (90% female participants).

**GECCC** offers a large and diverse *corpus* for grammatical error correction in Czech containing 83,058 sentences across four domains: native student essays, informal website discussions, Romani minority student essays, and non-native learner texts (*Náplava et al., 2022*). It features professionally annotated errors using a Czech-specific ERRANT adaptation, with error rates varying significantly by domain (5.8% in native formal writing to 26.2% in Romani essays). This multi-domain approach provides a comprehensive representation of error patterns across different Czech-speaking populations.

**BGEC** provides a large-scale GEC *corpus* for Bangla consisting of 7.07 million source-target pairs featuring ten distinct error types (*Hossain et al., 2024*). The dataset was created by augmenting text with grammatical errors (verb inflection, number, punctuation, missing elements, *etc.*), it produces diverse morphological and syntactic variations. The *corpus* supports a MarianMT-based model that achieves high performance (83.3% accuracy, 0.833 F1, 95.9 SacreBLEU). The main limitation lies in the reliance on synthetic rather than authentic writer data, which may restrict real-world application and domain coverage.

**Eo-GEC** comprises 307 grammatically incorrect sentences with corrections and fine-grained linguistic annotations (*Liang, 2024*). Sources include Plena Manlibro de Esperanta Gramatiko (45.3%), Teach Yourself Complete Esperanto (32.6%), Nivelo al Nivelo exam resources (12.1%) and authentic user writing (10.1%). With an average sentence length of 7.64 words, replacement errors (77.6%), missing errors (16.3%), and unnecessary errors (6.1%). Common error categories involve noun cases (19.7%), verb forms (18.0%), and table words (13.2%), reflecting key challenges in Esperanto's accusative system and correlative structures.

**GNC** contains 227 sentences from native Greek high school student essays, manually digitized from handwritten texts (*Korre, Chatzipanagiotou & Pavlopoulos, 2021*). It features fine-grained linguistic annotations categorizing grammatical errors with specific error types. Sentences may contain multiple errors: 44.9% error-free, 38.3% containing one error, 11.9% have two errors, and 4.8% contain three or more. Evaluation of ELERRANT on this dataset shows 77.30–83.91% accuracy, with high inter-annotator agreement

(84.65% Cohen's Kappa). Common errors include spelling and accentuation issues, reflecting typical challenges posed by Greek's complex morphological system.

**SPECIL** represents the first dedicated spelling error *corpus* for Indonesian, addressing a critical resource gap in NLP (*Yanfi et al., 2023*). It contains 21,500 sentences with over 180,000 tokens across three domains (Indonesian language, natural science, social science), it documents six error types: non-word errors (transposition, substitution, insertion, deletion), real-word errors, and punctuation errors. The *corpus* includes approximately 86,000 distinct words with non-word errors and 22,000 with real-word errors across domains, identified using a manually created dictionary of 843 words. This resource enables the development of Indonesian-specific spell-checking tools and language models.

**IceEC** comprises 4,044 texts from three genres: native Icelandic high school student essays, online news, and Wikipedia articles, with 56,794 categorized error instances across 44,268 revision spans (*Arnardóttir et al., 2021*). Error rates vary by genre: Wikipedia (62.03 errors/ 1,000 words), student essays (37.83), and news (35.74). Common error categories include punctuation (25.46%), wording (14.74%), and spacing (6.98%). The annotation process involved specialized annotators and employed a hierarchical scheme comprising six main categories, 31 subcategories, and 253 error codes. Published in augmented TEI-format XML under CC BY 4, IceEC addresses the need for error-annotated data in this low-resource language.

**RULEC-GEC** contains 12,480 sentences (206K words) from university essays written by foreign learners and speakers of heritage (*Rozovskaya & Roth, 2019*). With an overall error rate of 6.3%, it captures 23 error categories, primarily spelling (21.7%), noun case (13.2%) and lexical choice (12.3%). The *corpus* reveals distinct error patterns between foreign learners (6.9% error rate) and heritage speakers (4.0% error rate), with the latter predominantly making spelling and punctuation errors.

**ReLCo** takes an innovative approach, by collecting data from interactions with the Revita language learning system (*Katinskaia et al., 2022*). Comprising 15,568 sentences containing grammatical errors and 6,802 sentences with non-word errors, it captures student responses to fill-in-the-blank exercises rather than complete compositions. This methodology enables continuous data collection through learning activities, with automatic error detection supplemented by manual verification.

**RU-Lang8** offers the greatest linguistic diversity, extracted from an online language learning platform with users from various first-language backgrounds (*Trinh & Rozovskaya, 2021*). The *corpus* includes 48,260 sentences (633K tokens) with high error rates of 15.6% in development and 11.3% in test sets. Japanese speakers form the largest contributor group (38%), followed by English (14%) and Korean speakers. Most errors involve replacements (71-74%), with insertions and deletions comprising smaller proportions.

**Tibetan GEC** *Corpus* addresses genitive and ergative grammatical errors in Tibetan text, containing 1.79 M training sentence pairs and two test sets (5 and 1.15K sentences) (*Jiacuo et al., 2020*). The authors implemented a BERT+Bi-LSTM architecture to predict correct grammatical cases, achieving 98.38% and 86.16% accuracy on the test sets. This *corpus* focuses on Tibetan's complex auxiliary word system, where genitive is the sixth case and ergative is the third case according to Tibetan grammar theory.

**RONACC** is the first Romanian GEC dataset with 10,119 sentence pairs in three categories: sentences without verbs, well-formed sentences and sentences with written mistakes (*Cotet, Ruseti & Dascalu, 2020*). The *corpus* includes TV/radio content corrected by the National Audiovisual Council. The authors adapted ERRANT for Romanian evaluation and created a 10M artificial *corpus* from Wikipedia for pre-training. Their Transformer-base model pre-trained on synthetic data and fine-tuned on RONACC achieved $F_{0.5} = 53.76$, significantly outperforming their baseline ($F_{0.5} = 44.38$).

The Thai **UGWC** *corpus* presents a novel two-stage pipeline for Thai spelling correction and word normalization in social media texts (*Lertpiya, Chalothorn & Chuangsuwanich, 2020*). The dataset contains 15,576 sentences with six types of annotated errors: misspelled words (61.38%), morphed words (24.00%), abbreviations (15.00%), spoonerisms (0.14%), slang (0.08%), and other errors (5.63%). Their approach combines a bidirectional LSTM error detector with a sequence-to-sequence corrector featuring contextual attention, achieving a 2.07% word error rate and a 0.9502 GLEU score, significantly outperforming existing methods, including the copy-augmented Transformer (2.51% WER).

The **COWS-L2H** *corpus* provides parallel *corpus* of corrected data from Spanish learner data with 33,950 sentences from 737 university students (*Davidson et al., 2020*). The dataset includes error annotations and corrected text from both native and non-native speakers, enabling the development of grammatical error correction tools to support Spanish teaching communities.

The **UA-GEC** presents the first Ukrainian grammatical error *corpus* with 33,735 sentences from 828 contributors in diverse writing domains (*Syvokon et al., 2023*). Available in two versions (GEC+Fluency and GEC-only), it features professional annotations for grammar (14.4%), spelling (19%), punctuation (43%) and fluency errors (23.6%), establishing a baseline with an mBART-50-large model that achieves $F_{0.5}$ scores of 0.61–0.69.

Based on the compiled *corpus* descriptions, there is clear evidence that the quality of the annotation varies significantly between the datasets, primarily depending on the annotation strategy of each project and the expertise of its annotators. Several corpora, such as GNC (Greek), report high inter-annotator agreement (84.65% Cohen's Kappa), highlighting consistent and reliable labeling. Similarly, some Czech corpora (*e.g.*, AKCES-GEC, GECCC) rely on professional annotators and well-defined annotation schemes (M2 format, ERRANT adaptations), suggesting a stronger foundation for robust grammatical error correction (GEC) evaluations. In contrast, community-annotated resources (*e.g.*, certain large-scale Russian data from language learning platforms) may encounter inconsistencies arising from volunteer contributors with diverse levels of

linguistic expertise. Synthetic datasets such as BGEC (Bangla) and partially synthetic resources (*e.g.*, Tibetan GEC *Corpus*, RONACC's extended Wikipedia-based data) offer large volumes of training material but risk introducing annotation noise unless rigorously verified. Hence, while most projects acknowledge the importance of data consistency, in practice they exhibit varying degrees of quality control, ranging from strictly curated professional annotations to open community-sourced edits.

In terms of error coverage, the corpora span a wide spectrum of linguistic phenomena. Some focus on extensive morphological challenges: the Tibetan GEC *Corpus* highlights genitive/ergative forms, while Eo-GEC (Esperanto) emphasizes case marking and verb forms. Others, such as QALB (Arabic) and RULEC-GEC (Russian), categorize errors by major action types (edits, insertions, deletions) or lexical components (spelling, lexical choice). There are also corpora that adopt a richly layered approach, including ICEEC (Icelandic) and GECCC (Czech), which distribute errors into numerous subcategories (more than 200 in some instances) and distinguish domains such as academic writing, online text, minority language varieties, and learner essays. This detailed stratification helps capture a broad range of real-world errors from punctuation and spelling to more intricate grammatical or idiomatic errors and identifies common points of difficulty for learners and native speakers. Consequently, while each dataset addresses at least a subset of core error types relevant to its target language, their overall coverage of common errors differs substantially based on the linguistic structures prioritized by researchers and the domains or user populations under study.

## CROSS-LINGUISTIC PERFORMANCE IN REAL-WORLD APPLICATION

Recent developments in GEC have placed increasing emphasis on practical implementation, resulting in a range of systems evaluated in diverse real-world settings, as shown in Table 6. One such example is ALLECS, a lightweight web-based GEC platform designed for low-resource environments and general users. Rather than proposing algorithmic improvements in isolation, it integrates multiple state-of-the-art models based on both generation and tagging and combines them through edit-based and text-based fusion techniques (*Qorib, Moon & Ng, 2023*). Its strength lies in its ability to maintain responsive correction performance under infrastructure constraints such as limited bandwidth and mobile devices, making it suitable for English learners as a second or foreign language.

GEC for spoken input presents an additional challenge. A recent end-to-end approach based on the Whisper automatic speech recognition (ASR) model has been evaluated for real-world oral language assessments such as the Linguaskill test (*Bannò et al., 2024*). This work addresses issues such as disfluency detection, the lack of annotated training data for spoken errors, and the need to provide accurate and usable feedback to learners during speaking practice. While performance is constrained by data availability, evaluation using traditional metrics—such as Precision, Recall, and $F_{0.5}$—demonstrates the system's potential for educational feedback in speech-based interfaces.

**Table 6 Comparison of grammatical error correction systems and research across languages.**

| Lang | System | Domain | Method/Architecture | Evaluation dataset/ Setting | Real-world focus | Findings |
|---|---|---|---|---|---|---|
| [EN] | ALLECS | General/ public users | Integration of 3 SOTA models (generation and tagging) | Internally tested; optimized for mobile and web | Lightweight deployment; ESL/EFL learners | Designed for low-bandwidth settings; accessible on mobile and web without performance sacrifice |
| [EN] | End-to-End Spoken GEC (Whisper-based) | Spoken, L2 context | Whisper ASR + end-to-end GEC pipeline | Linguaskill (spoken exam data) | Voice-based feedback for learners | Evaluated using Precision, Recall, $F_{0.5}$; challenges in disfluency and limited annotated data |
| [EN] | Learner Writing GEC (Seq2Seq + Transformer) | Student essays | Transformer, Seq2Seq, integrated scoring | ICNALE, Brown *Corpus* | Autonomous student feedback systems | Applied in real settings; uses $F_{0.5}$-score to prioritize precision in writing correction |
| [CN] | NaSGEC (Chinese GEC dataset) | Native speaker writing | Benchmarked using SOTA CGEC models | Multi-domain: social media, exams, scientific text | Writing aids and proofreading tools | Performance declines when trained on general corpora and applied across domains |

Cross-linguistic and domain-specific variations in real-world GEC performance are also prominent. In Chinese, the introduction of the NaSGEC *corpus* shifted focus toward domain-sensitive error correction by including native speaker data across multiple domains: social media, academic writing, and standardized assessments (*Zhang et al., 2023*). Experimental results on this dataset show clear drops in model performance when systems trained on general corpora are applied to more specialized or informal domains, highlighting the persistent challenge of domain adaptation.

*Cheng & Qiao (2022)* introduce a deep neural GEC framework that fuses Bidirectional RNN with linguistically informed characteristics, replacing baselines based on rules and n-gram while retaining language-independent design principles. Trained on an English learner *corpus*, the model attains superior precision and recall. The authors contend that, owing to its reliance on universal part-of-speech tags and semantic cues, the architecture can be effectively transferred to typologically diverse languages with only minimal re-tuning, thereby demonstrating significant potential for cross-linguistic generalization. They also describe its integration into an intelligent writing evaluation platform, where the network provides real-time feedback to thousands of users, confirming robustness outside laboratory settings. These results underscore the ability of the framework to sustain high performance in languages and in large-scale real-world educational applications.

# GAPS IN THE LITERATURE AND CHALLENGES

## Overlooked aspects

Despite advancements in GEC for low-resource languages, several important aspects remain underexplored:

**Limited and inconsistent focus on linguistic diversity**. Most existing studies on linguistic diversity primarily concentrate on a narrow set of morphologically rich languages, such as Arabic (*Mahmoud et al., 2023*; *Alhafni et al., 2023*), Icelandic (*Ingólfsdóttir et al., 2023*),

and Persian (*Golizadeh, Golizadeh & Forouzanfard, 2022*). Research on other typologically diverse languages, including those with free-word order, agglutinative structures, or polysynthetic tendencies, is limited. Zarma (*Keita et al., 2024*) and Tagalog (*Lin et al., 2023*) represent rare additions to this space, but are not sufficient to capture the full spectrum of linguistic diversity. Many multilingual GEC articles emphasize scalability, but do not adequately address typologically distant languages, leading to suboptimal adaptation for languages that differ significantly from high-resource languages such as English (*Sun et al., 2022*; *Yamashita et al., 2020*; *Luhtaru, Korotkova & Fishel, 2024*).

**Challenges in adapting multilingual models**. Multilingual models (*e.g.*, mT5, PXLM) often struggle with negative transfer when typologically unrelated languages are jointly trained (*Sun et al., 2022*; *Yamashita et al., 2020*). For example, languages with rich morphology or complex syntactic structures often interfere with simpler languages during training, resulting in degraded performance. These models also tend to be tuned for tasks that favor frequent and well-documented languages, further marginalizing extremely low-resource languages with unique linguistic features (*Luhtaru, Korotkova & Fishel, 2024*; *Korre & Pavlopoulos, 2022*).

## Evaluation and metrics gaps
Current evaluation practices encounter significant challenges in handling syntactically and morphologically diverse low-resource languages:

**Limited exploration of custom metrics**. Standard metrics such as BLEU, GLEU, and ERRANT are widely used in evaluation benchmarks, but are insufficient for highly polymorphic or syntactically complex languages. For example, the adaptation of ERRANT for Greek (*Korre & Pavlopoulos, 2022*) acknowledges such gaps, but remains rare, and similar efforts are lacking for languages such as Zarma or agglutinative languages (*Ingólfsdóttir et al., 2023*; *Keita et al., 2024*). There has been little exploration of how to develop metrics that accommodate errors unique to typologically diverse languages, such as those involving agreement, case marking, or compounding.

**Reliance on standard metric-based evaluation**. Studies frequently adopt existing metrics without tailoring them to reflect low-resource or culturally specific nuances (*Sun et al., 2022*; *Flachs, Stahlberg & Kumar, 2021*; *White & Rozovskaya, 2020*). This results in evaluations that can overlook or penalize legitimate linguistic variations that deviate from the norms of high-resource benchmarks like W&I+LOCNESS or CoNLL14 (*Grundkiewicz, Junczys-Dowmunt & Heafield, 2019*). Weakly supervised evaluation methods, needed for extremely low-resource languages with minimal annotated corpora, are largely absent from the literature (*Alhafni et al., 2023*; *Keita et al., 2024*).

## Resource limitations
The availability, standardization and accessibility of datasets for low-resource GEC remain critical bottlenecks, which hinders consistent progress.

**Need for more standardized datasets**. While curated datasets exist for specific languages (*e.g.*, Zarma (*Keita et al., 2024*), Indonesian (*Lin et al., 2025*; *Musyafa et al., 2022*), Arabic (*Mahmoud et al., 2023*; *Alhafni et al., 2023*)), their methodologies vary widely, complicating cross-comparison. There remains a critical lack of datasets that represent languages spoken in Africa, Southeast Asia, and among indigenous populations (*Golizadeh, Golizadeh & Forouzanfard, 2022*; *Keita et al., 2024*; *Flachs, Stahlberg & Kumar, 2021*). Additionally, typologically unique languages, including polysynthetic and tonal varieties are virtually excluded from current datasets, leaving entire linguistic categories unaddressed.

# DIRECTIONS FOR FUTURE RESEARCH

Based on our comprehensive review of GEC approaches for low-resource languages, several promising research directions emerge. These recommendations address the fundamental challenges of data scarcity, linguistic diversity, computational efficiency, and evaluation frameworks.

## Enhancing synthetic data generation techniques

Future research should focus on developing more sophisticated synthetic data generation methods that accurately reflect language-specific error patterns. Advanced approaches could include the following: Linguistically informed adversarial techniques, as demonstrated with GAN-generated errors for Persian (*Golizadeh, Golizadeh & Forouzanfard, 2022*), to create highly contextualized and realistic error patterns; and developing typologically informed noising methods that align with the linguistic characteristics of low-resource languages, such as case marking errors or syntactic reordering for free word order languages (*White & Rozovskaya, 2020*).

These advancements would improve both the realism and diversity of synthetic corpora, thereby enhancing the robustness of the model across a wider range of error types and languages.

## Advancing multilingual and cross-lingual approaches

Expanding the capabilities of multilingual models for typologically diverse languages requires addressing several challenges. One key challenge is developing strategies to mitigate negative transfer between unrelated languages, such as language clustering (training similar languages together) or weight balancing of language during multitask learning (*Sun et al., 2022*; *Luhtaru, Korotkova & Fishel, 2024*). This include creating parameter-efficient adaptation methods such as adapters, LoRA (Low-Rank Adaptation), or prefix tuning that optimize performance for specific low-resource languages without fine-tuning the entire model (*Sun et al., 2022*), and investigating meta-learning frameworks that enable rapid adaptation to new languages with minimal annotated data.

These approaches would significantly improve knowledge transfer across language boundaries while respecting typological differences, making them computationally efficient and well aligned with the data-sparse context of low-resource GEC.

## Innovations in tokenization and embeddings

Current embedding models such as mBERT or XLM-R struggle with the nuances of morphologically complex and typologically diverse languages. Future work should focus on the following directions: Incorporate morphology-aware embeddings, where subword or byte-level tokenization respects agglutination, compounding, and inflections commonly found in low-resource languages (*Ingólfsdóttir et al., 2023*; *Flachs, Stahlberg & Kumar, 2021*). Explore language-specific embedding modules that reflect features such as tone marking, grammatical agreement, or compounding, drawing from insights into languages such as Icelandic (*Ingólfsdóttir et al., 2023*) or Indonesian (*Musyafa et al., 2022*). Future systems should adopt more flexible architectures that can better accommodate diverse linguistic features without requiring extensive language-specific engineering. These innovations would bridge representation gaps and ensure that GEC systems better capture language-specific structural errors.

## Expanding linguistic and stylistic adaptations

The growing prevalence of code-switching (CSW), particularly in multilingual regions, necessitates further exploration of GEC techniques: These include developing GEC techniques tailored for CSW input that incorporate linguistic diversity at language-switching points, as well as pragmatic considerations in multilingual texts (*Potter & Yuan, 2024*). Researchers should also refine data augmentation methods to better match the stylistic nuances of learner texts, such as ensuring that non-native translationese data (*Fang et al., 2023a*) align with the writing styles of GEC target corpora. Additionally, it is essential to create dynamic models that can handle the transition between different grammatical systems within the same text.

## Dataset development

To expand low-resource GEC to underrepresented linguistic groups, dataset creation should prioritize: Regionally significant language families, including African languages (*e.g.*, Bantu, Niger-Congo) and Pacific-based languages (*e.g.*, Austronesian) (*Keita et al., 2024*). Improving annotation approaches through crowd-sourcing with native speakers or LLM-assisted annotation (*e.g.*, GPT-assisted error identification), as shown in Indonesian *corpus* development (*Lin et al., 2025*). Developing family-specific approaches that take advantage of shared linguistic characteristics while respecting individual language differences.

## Adaptation of evaluation frameworks

Current metrics do not adequately handle morphological and syntactic complexity. Future research should focus on: (1) Tailoring evaluation frameworks to reflect typological nuances, including morphological adaptations for languages with extensive case or agreement systems (*e.g.*, ERRANT for Greek (*Korre & Pavlopoulos, 2022*)). (2) Developing typology-based evaluation modules that extend standard metrics such as GLEU to handle

compounding, word order, and semantic roles specific to low-resource languages.
(3) Creating multi-reference benchmarks that acknowledge the multiple valid corrections often possible in morphologically rich languages (*Rozovskaya & Roth, 2021*).
(4) Establishing evaluation approaches that balance grammatical accuracy with semantic preservation.

### Computational efficiency and accessibility

For practical deployment in low-resource settings, future research should prioritize the following lightweight model architectures optimized for computational efficiency. Knowledge distillation from large multilingual models to more compact, language-specific systems. Hybrid approaches that intelligently combine rule-based components with neural architectures to reduce computational demands. Cutting-edge LLMs, while powerful, should be adapted through techniques such as quantization and pruning to function effectively in bandwidth-constrained environments. Real-world implementations should prioritize offline capabilities and progressive loading techniques to ensure that GEC technologies remain accessible in regions with intermittent connectivity or limited computing resources, democratizing access to writing support tools regardless of technological infrastructure.

By addressing these research directions, the field can move toward more inclusive, effective, and accessible grammatical error correction systems that serve the global linguistic landscape in all its diversity.

## CONCLUSION

This literature review synthesizes current research on GEC for low-resource languages, highlighting effective strategies to overcome key challenges in this emerging field. Our analysis reveals that data scarcity remains the most significant barrier to developing robust GEC systems for typologically diverse languages.

The most promising approaches combine multiple complementary strategies: synthetic data generation techniques to address limited annotated corpora, cross-lingual knowledge transfer to leverage high-resource language models, and hybrid architectures to integrate linguistic knowledge with neural approaches.

Our review identifies several important gaps in current research, particularly in evaluation frameworks for linguistically diverse languages, dataset development for underrepresented language families, and computational approaches tailored to typological diversity. Addressing these gaps will require greater interdisciplinary collaboration between computational linguists and native speakers of low-resource languages.

As the field advances, we expect more inclusive GEC systems that can effectively serve the global linguistic landscape beyond the few high-resource languages which currently dominate research. This expansion will play a crucial role in advancing language education, digital communication, and content creation across the world's diverse languages.

### Funding
This work was supported by the National Natural Science Foundation of China under Grant Nos. 62476247, 62072409, and 62176234. The funders had no role in study design, data collection and analysis, decision to publish, or preparation of the manuscript.

### Grant Disclosures
The following grant information was disclosed by the authors:
National Natural Science Foundation of China: 62476247, 62072409, and 62176234.

### Competing Interests
Xiangjie Kong is an Academic Editor for PeerJ.

### Author Contributions
- Syauqie Muhammad Marier conceived and designed the experiments, performed the experiments, analyzed the data, performed the computation work, prepared figures and/or tables, authored or reviewed drafts of the article, and approved the final draft.
- Xiangfan Chen performed the computation work, prepared figures and/or tables, and approved the final draft.
- Linan Zhu conceived and designed the experiments, performed the experiments, analyzed the data, authored or reviewed drafts of the article, and approved the final draft.
- Xiangjie Kong conceived and designed the experiments, performed the experiments, analyzed the data, authored or reviewed drafts of the article, and approved the final draft.

### Data Availability
This is a literature review.

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
