# Peer review of "Grammatical error correction for low-resource languages: a review of challenges, strategies, computational and future directions"

_PeerJ Computer Science, doi:10.7717/peerj-cs.3044_

## Round 0.1 · original submission · Major Revisions

Dear Authors,

Thank you for submitting your Literature Review article. Reviewers have now commented on your article and suggest major revisions. We do encourage you to address the concerns and criticisms of the reviewers and resubmit your article once you have updated it accordingly. When submitting the revised version of your article, it will be better to also address the following:

1. It is imperative that errors pertaining to English grammar and writing style are rectified.
2. “In ?? , we explore detail each of these techniques.” should be corrected.
3. “In ?? , we present a computational techniques:” should be corrected.
4. “These challenges are compounded by the scarcity of high-quality datasets or language processing tools, as most open-source NLP pipelines are tailored to resource-rich languages like English Sun et al. (2022); Luhtaru et al. (2024a); Musyafa et al. (2022).” etc should be corrected.

Best wishes,


**Language Note:** The review process has identified that the English language must be improved. PeerJ can provide language editing services - please contact us at [email protected] for pricing (be sure to provide your manuscript number and title). Alternatively, you should make your own arrangements to improve the language quality and provide details in your response letter. – PeerJ Staff

Reviewer 1 ·

Basic reporting

This review consolidates insights from essential studies to showcase effective GEC methods for low-resource languages. It emphasizes innovative approaches that tackle the challenges of limited annotated corpora, typological complexities, and evaluation difficulties, making a compelling case for advancing research in this critical area.

1- In line 53, "We focus on answering the following questions: 'Which strategies prove to be the most effective for GEC in languages that have limited resources, and how can these strategies be tailored to effectively tackle the challenges of data scarcity and linguistic diversity?' should be stated clearly in the contribution side.
2- To enhance clarity and understanding, I recommend moving Figure 1 to the background and the Key Concepts section. This adjustment will allow for a more detailed explanation, ensuring that its significance is fully conveyed.

Experimental design

The study has been executed with great attention to detail. A comprehensive study design, complemented by relevant studies and figures, has been presented. However, the abundance of subheadings has created interruptions in the study's flow. By consolidating key subheadings and expanding on them under more general headings, we can enhance clarity and improve the overall coherence of the study.

Validity of the findings

This study Current comparisons of cutting-edge approaches reveal that nnovative methods that harness few-shot learning and fine-tuned Large Language Models (LLMs) pave the way for more scalable and adaptable GEC solutions, promising significant advancements in the field. it is essential to enhance computational efficiency ensuring that solutions are accessible and effective in various contexts, including those with limited technological capabilities. This study will ultimately contribute to more robust and inclusive language technologies.

Additional comments

Titles can be effectively organized into broader categories, which not only helps in creating a logical framework but also allows for easier navigation through multiple topics. By adopting a more general article format, we can ensure that each category is clearly defined and that readers can quickly grasp the main ideas presented. This approach not only enhances the clarity of the information but also increases engagement by presenting content in a more structured and reader-friendly manner.

It's important to provide a detailed explanation of key figures, such as Figure 1, within the background and Key Concepts section to enhance understanding and clarity.


Article contributions should be presented in a more comprehensive and detailed manner at the end of the introduction. The use of numerous subheadings based on categorization can be reduced, and the language should be more general and focused on contributions.

Reviewer 2 ·

Basic reporting

The article contains many spelling and grammar errors. There are consecutively repeated words (Figure Figure 2, etc.). In general, the article needs to be reviewed carefully.

There are many omissions or errors in the form of citations, such as spaces, syntax and spelling errors.

Various studies have been cited (e.g., Belkebir and Habash (2021); Uz and Eryigit (2023); Korre et al. (2021). However, no summary is given about these studies. Brief explanations about how the studies address the subject and their shortcomings would make the article stronger.

Experimental design

The paper discusses many theoretical approaches, but does not provide sufficient data on experimental results and real-world applications of these approaches. More examples and performance comparisons could be added, especially on the results of experiments conducted in different languages.

There is a lack of detail for Hybrid Rule-Based + Neural Approaches. The advantages offered by these approaches should be supported by cited studies. With sufficient detail, it should be clearer why the models are chosen and what problems they solve.

The terms (agglutinative languages, free word order, etc.) are explained superficially in the article. More detailed explanations will enable the reader to understand these linguistic concepts better. In addition, the relationship between these concepts and GEC systems should be stated more clearly.

There is no sufficient architectural detail given about tools like ARETA, ELERRANT etc. It is not clearly stated which methods or resources they use.

The information provided about the datasets is unbalanced. While some datasets are described in detail (e.g. AKCES-GEC), only one or two sentences are given about others (e.g. SPECIL).

Datasets are given, but no critical analysis has been made about them, such as which ones have low labeling quality and which ones cover the most common errors.

Validity of the findings

The conclusion section contains too many technical details. It should be simpler, clearer and more conclusion-oriented.

---

## Round 0.2 · Minor Revisions

Dear Authors,

Thank you for the revised manuscript. One new reviewer suggests minor revision. We encourage you to address the concerns and criticisms of Reviewer 1 and resubmit your paper once you have updated it accordingly.

Best wishes,

Reviewer 3 ·

Basic reporting

The manuscript is related to a review on gramatical error corrections in low-source langeuages. The main topics of the paper are synthetic data generation, multilingual pretraining, cross-lingual transfer learning, and typology-aware approaches. It also includes the comparative evaluations and analysis of models, evaluation metrics, and datasets. The paper is well-structured wiht sections such as introduction, methodlogy, key challanges, and approaches. However, the tracked changes version reveals numerous grammatical and stylistic issues (e.g., inconsistencies in verb tense, sentence fragments, awkward phrasing).A final proofread is necessary to ensure professional readability and fluency. Some grammatical errors are given below.
- “The second discusses methods for handling data scarcity…”
Correction: “The second section discusses methods for handling data scarcity…”
- “…particularly to low-resource languages.”
Correction: “…particularly for low-resource languages.”
- Duplication research papers are published in different journals.”
Correction: “Duplicate research papers published in different journals.”

Experimental design

The content of tha manuscript is related to grammatical error correction. For this purpose, different approaches such as LLM, transformer-based and multi-lingual pretrained models are investigated. As a survey paper, it does not include new human/animal research data, and ethical compliance is not directly relevant. All cited works are appropriately referenced, and the authors' affiliations are transparent.Figures like the taxonomy of GEC approaches (Figure 2) and data augmentation methods (Figure 3) effectively visualize complex information.
The paper provides a solid background on low-resource languages, their linguistic challenges, and GEC systems.In the paper, recent studies have been cited and related dataset are mentioned.

Validity of the findings

The manuscript addresses an important gap, is technically sound, and presents practical insights. Minor grammatical polishing is necessary to enhance clarity and professional tone.

---

## Round 0.3 · accepted · Accept

Dear Authors,

Reviewers are satisfied that your paper is acceptable and suitable for publication.

Best wishes,

Reviewer 2 ·

Basic reporting

Spelling and grammatical errors have been corrected, repetitive expressions and language errors have been eliminated, and suggested improvements for citations and sources have been taken into account.

Experimental design

The experimental data and information on real-world applications added to the theoretical approaches are very useful. The experimental results and comparisons in different languages ​​have increased the practical value of the paper. The details you added about hybrid rule-based + neural approaches are quite satisfactory. The additional explanations you provided about linguistic concepts are appropriate and understandable. The architectural information you added about tools such as ARETA and ELERRANT has better clarified the working principles of these tools and the methods they use. The arrangements you made about the datasets are quite comprehensive.

Validity of the findings

Conclusions are presented in a clearer and more understandable way by simplifying and reducing technical details.

Reviewer 3 ·

Basic reporting

Based on the detailed response letter and the tracked changes in the manuscript, the authors have sufficiently addressed the reviewer’s comments and performed the necessary revisions, including grammatical corrections and structural improvements.

Experimental design

No correction has been given regarding this section.

Validity of the findings

No correction has been given regarding this section.